# BETs inhibition attenuates oxidative stress and preserves muscle integrity in Duchenne muscular dystrophy

Marco Segatto[1,2,5], Roberta Szokoll[1,5], Raffaella Fittipaldi[1], Cinzia Bottino[1], Lorenzo Nevi [1], Kamel Mamchaoui[3], Panagis Filippakopoulos[4] & Giuseppina Caretti [1✉]

Duchenne muscular dystrophy (DMD) affects 1 in 3500 live male births. To date, there is no effective cure for DMD, and the identification of novel molecular targets involved in disease progression is important to design more effective treatments and therapies to alleviate DMD symptoms. Here, we show that protein levels of the Bromodomain and extra-terminal domain (BET) protein BRD4 are significantly increased in the muscle of the mouse model of DMD, the mdx mouse, and that pharmacological inhibition of the BET proteins has a beneficial outcome, tempering oxidative stress and muscle damage. Alterations in reactive oxygen species (ROS) metabolism are an early event in DMD onset and they are tightly linked to inflammation, fibrosis, and necrosis in skeletal muscle. By restoring ROS metabolism, BET inhibition ameliorates these hallmarks of the dystrophic muscle, translating to a beneficial effect on muscle function. BRD4 direct association to chromatin regulatory regions of the NADPH oxidase subunits increases in the mdx muscle and JQ1 administration reduces BRD4 and BRD2 recruitment at these regions. JQ1 treatment reduces NADPH subunit transcript levels in mdx muscles, isolated myofibers and DMD immortalized myoblasts. Our data highlight novel functions of the BET proteins in dystrophic skeletal muscle and suggest that BET inhibitors may ameliorate the pathophysiology of DMD.

[1] Department of Biosciences, Università degli Studi di Milano, Via Celoria 26, 20133 Milan, Italy. [2] Department of Biosciences and Territory, University of Molise, Contrada Fonte Lappone, Pesche (Is), Italy. [3] Sorbonne Université, Inserm, Institut de Myologie, U974, Center for Research in Myology, 47 Boulevard de l'hôpital, 75013 Paris, France. [4] Structural Genomics Consortium, Old Road Campus Research Building, Nuffield Department of Medicine, Oxford OX3 7DQ, UK. [5] These authors contributed equally: Marco Segatto, Roberta Szokoll. ✉email: giuseppina.caretti@unimi.it

D uchenne muscular dystrophy (DMD) is the most common form of muscular dystrophy. This X-linked recessive disorder is caused by mutations in the dystrophin gene, and it affects approximately 1 in 3500 male births worldwide[1].

A decisive therapy for DMD treatment is not available yet. Genome editing approaches hold extensive promise for a future resolutive strategy to pursue[2], but current clinical approaches are still not effective in reversing the phenotype. Thus, pre-clinical and clinical studies are focusing on pharmacological therapies targeting downstream events of the genetic mutation, including inflammation, fibrosis, adipocyte infiltration, and metabolism[3].

Dystrophin is a large structural protein located at the sarcolemma that mechanically links the internal cytoskeleton to the extracellular matrix, thus conferring membrane stability during contraction[4]. Lack of dystrophin dramatically increases the sarcolemma susceptibility to contraction-induced injury, leading to myofiber necrosis and triggering secondary events, such as inflammation and fibrosis[5].

The direct molecular mechanisms for loss of muscle function in mouse models and DMD patients are still under active investigation. Several aberrant processes (e.g., intracellular calcium homeostasis, inflammation and ROS metabolism) are indeed implicated as early events in the disease pathophysiology[6–8], since they result in the activation of calcium-dependent degradative pathways, myofibrils damage and necrosis, incomplete regeneration cycles, autophagy impairment, increased fibrosis and adipose tissue accumulation[9–13]. In particular, muscle biopsies from DMD patients show increased oxidative stress compared to controls[14,15] and increased NADPH oxidase (Nox2) activity as an early event in the disease onset[7,16], preceding immune cells infiltration and necrosis[7]. Increased oxidative stress has been recently causally linked to autophagy impairment in the mdx dystrophic muscle[13] and the genetic elimination of Nox2-mediated ROS production has been reported to reduce inflammation and fibrosis[13,17].

Gene expression profiles of skeletal muscle are altered in muscular dystrophies[18–20] and the epigenetic regulation of muscle stem cells plays a crucial role for their regenerative potential in the mdx model[21–25]. In addition, dystrophin loss leads to alteration in signaling pathways that eventually translate in transcriptional reprogramming. For example, histone-deacetylase (HDAC) activity is perturbed by dystrophin deficiency and this contributes to transcriptional alteration in mdx mice[26,27]. Furthermore, epigenetic drugs targeting HDACs are in clinical trial for DMD and showed promising results in the histological progression of the disease[28–30].

We have recently shown that the BET protein BRD4 promotes muscle atrophy in an in vitro model of glucocorticoid-induced atrophy and in experimental models of cancer cachexia[31,32]. Because of this evidence and the well-established role played by BRD4 in inflammation[33–35], we aimed to characterize BRD4 contribution in skeletal muscle pathophysiology of a mouse model of DMD. In this study, we show that BRD4 influences ROS metabolism by regulating the transcriptional activation of different subunits of the NADPH oxidase complex in the mdx muscle. Furthermore, administration of the BET inhibitor JQ1 reduces oxidative stress and ameliorates skeletal muscle morphology and muscle function. JQ1 treatment rescues autophagy and dramatically restricts muscle damage, preventing muscle inflammation and fibrosis, and tempering muscle regeneration.

## Results

**BRD4 levels increase in the muscle of DMD patients and of the mdx mouse.** The involvement of BET proteins in inflammatory processes and in skeletal muscle homeostasis[31–34] prompted us to study their role in the mdx muscle. First, we examined their protein levels in tibialis anterior (TA) of control and mdx mice. BRD2 and BRD3 abundance was comparable in muscles from control and mdx mice, whereas BRD4 protein levels were significantly increased in the dystrophic muscle (Fig. 1a). Notably, the BRD4 antibody specificity was ascertained by silencing experiments (Supplementary Fig. 1A). Conversely, BRD2/3/4 transcripts were expressed at a similar rate both in control and mdx TAs, suggesting that post-transcriptional events are involved in BRD4 regulation (Fig. 1b). We next analyzed BRD4 levels in DMD muscle samples, and found that BRD4 protein was higher in muscles of DMD patients than in age-matched controls (Fig. 1c). We, therefore, interrogated RNA-Seq results published by Khairallah et al.[36] and found that BET transcript levels do not significantly change in DMD muscles (Fig. 1d).

Overall, these data show that BRD4 levels are higher in the muscle of Duchenne patients and in the mdx skeletal muscle, prompting us to further characterize BRD4 function in the mouse model for DMD.

**JQ1 treatment reduces muscle damage in the mdx mouse.** Based on our initial data (Fig. 1) and our previous observation that BRD4 blockade ameliorates glucocorticoid-induced atrophy in C2C12 myotubes, as well as taking into account the well-documented anti-inflammatory effect of BET inhibitors[33,35] we hypothesized that JQ1 treatment may ameliorate the dystrophic phenotype in the mdx mouse model of DMD.

We daily treated 10-week-old mdx mice with JQ1 (20 mg/kg per day) by intraperitoneal injection for two weeks and performed morphological studies on TA muscle sections to examine the effects of BET inhibition on the histopathology of dystrophic mdx muscle fibers. Hematoxylin/eosin staining confirmed the presence of a distinctive pattern of dystrophic muscle pathology in vehicle-treated mdx mice, evidenced by mononuclear cell infiltration and centrally located nuclei. Conversely, the number of infiltrating inflammatory cells was reduced in muscles of JQ1-treated mdx mice (Fig. 2a). In addition, cellular membrane permeability and subsequent fiber necrosis was reduced by JQ1 treatment, as revealed by the decreased number of Evans blue positive fluorescent cells (Fig. 2c). While the total number of fibers per area was identical in the mdx and JQ1-treated mdx muscles (Supplementary Fig. 2A), JQ1 administration significantly increased the number of peripherally nucleated fibers and decreased the centrally nucleated fibers (Fig. 2b), further suggesting that muscles from JQ1-treated mdx mice were less vulnerable to mechanical stress. The increased resistance to the dystrophic phenotype was also associated with reduced cell death, as revealed by the decreased levels of cleaved caspase-3 (Fig. 2d). As previously reported[37], succinate dehydrogenase (SDH) staining decreased in the mdx muscle compared to WT; however the staining intensity was recovered to that of the WT in JQ1-treated mdx mice (Fig. 2e and Supplementary Fig. 2B), indicating an improved energy metabolism following JQ1 administration, in the mdx TA muscle. Overall these data hint for a beneficial effect of JQ1 treatment in skeletal muscle of mdx mice.

**JQ1 restores autophagy in the mdx muscle.** To elucidate the molecular mechanisms underlying JQ1 beneficial effects in the mdx skeletal muscle, we investigated JQ1 impact on key metabolic processes, which are affected by the mdx physiopathology. Recent reports revealed that autophagy suppression contributes to the symptomatology of different forms of muscular dystrophies and is detrimental for the maintenance of muscle homeostasis[12,38]. To test whether autophagy recovery was, at least in part, responsible for JQ1-mediated amelioration in muscle

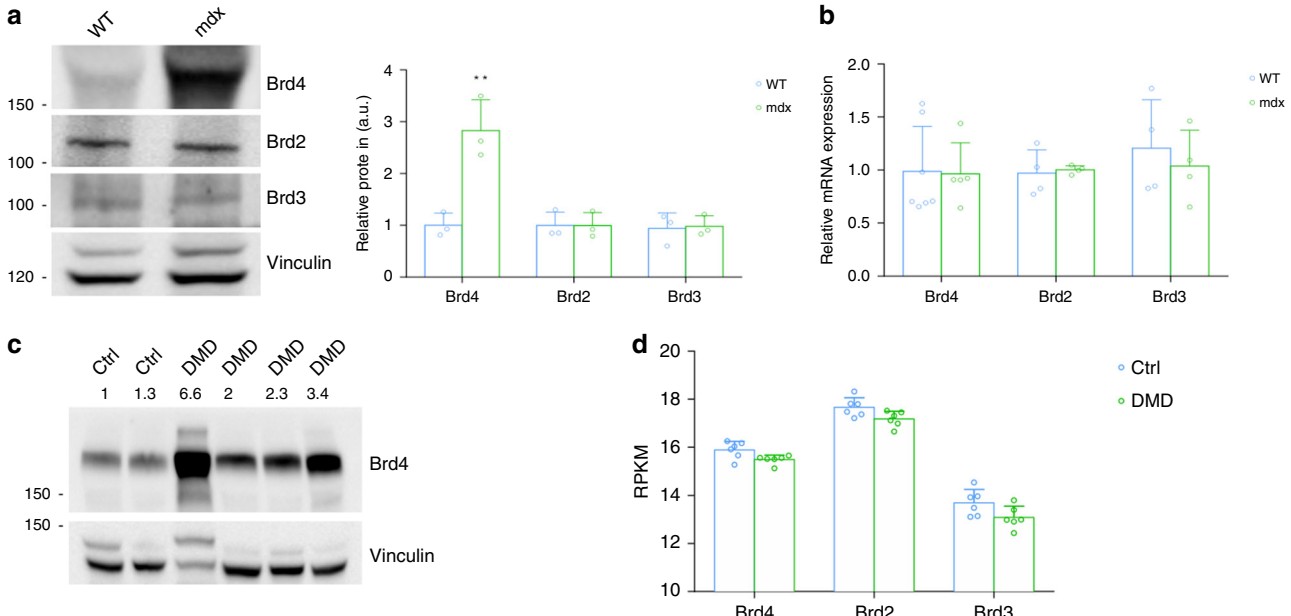

**Fig. 1 BRD4 protein levels are higher in DMD and mdx muscles. a** Representative images for immunoblot of BRD2, BRD3, and BRD4 in WT and mdx mice. Vinculin is used as a loading control. Right panel, signal quantification was performed with ImageJ. Data are expressed as the mean ± SD, $n = 3$, **denotes $P < 0.01$ was determined by using unpaired two-sided $t$-test. a indicates statistical significance compared to the control group. **b** RNA of control and mdx mice was analyzed for BRD2 (WT, $n = 4$; mdx, $n = 4$), BRD3 (WT, $n = 4$; mdx, $n = 4$) and BRD4 (WT, $n = 7$; mdx, $n = 5$) levels by qRT-PCR. Data were normalized against HPRT. Data are expressed as the mean ± SD, statistical significance was calculated using unpaired two-sided $t$-test. **c** Immunoblot analysis of paravertebral muscle specimen from Duchenne patients ($n = 4$) and healthy controls ($n = 2$). Normalized band intensity in immunoblots is reported above signals. **d** RPKM expression levels of BRD4 transcript in previously reported RNA-Seq dataset for DMD ($n = 6$) and healthy donors ($n = 6$). Data are expressed as the mean ± SD.

morphology, we examined the abundance of proteins involved in autophagy/lysosomal pathways and tested whether they were influenced by JQ1 treatment. As demonstrated by other reports[11,12,39], we found that the ratio between LC3II (the active lipidated LC3 form) and LC3I (the cytosolic inactive LC3 form) was reduced in TA muscles from vehicle-treated mdx mice. JQ1 treatment restored LC3II/LC3I ratios to levels comparable to the ones observed in control animals. Concurrently, p62 levels, which were upregulated in the mdx muscle, were comparable to those of control mice in TAs of JQ1-treated mdx mice (Fig. 3a). We concluded that JQ1 treatment promotes restoration of autophagy in the mdx mouse model. The rescue in autophagy could not be explained by a JQ1-mediated effect on transcriptional regulation of autophagy genes, because mRNA levels of a group of key autophagy genes did not increase following JQ1 treatment (Supplementary Fig. 3A). We, therefore, interrogated the activation state of different signaling pathways known to regulate autophagy. The metabolic sensor AMPK is a potent inducer of autophagy and its activation is known to decrease in mdx muscles when compared to control animals[39–41]. We observed that JQ1 administration was able to fully restore p-AMPK (Ser172) phosphorylation in mdx TAs. Likewise, AMPK-dependent phosphorylation of Ulk1 (Ser555), which plays a crucial role in autophagy initiation, was also upregulated in TAs from JQ1-treated mdx mice (Fig. 3b). In agreement with previous reports, AMPK phosphorylation correlated with Sirt1 protein levels[42,43], which were dramatically decreased in muscles from mdx mice. Sirt1 protein levels were recovered following JQ1 treatment. Consistent with this evidence, acetylation on histone H3 lysine 9 (H3-K9Ac) was upregulated in muscles from mdx mice and decreased following JQ1 administration (Fig. 3c).

Consistent with a rescue in autophagy, JQ1 treatment also mitigated the upregulation of Akt/mTOR/p70S6k pathway in the mdx muscle, and it led to activation levels for these kinases comparable to the ones observed in control animals (Fig. 3d).

Oxidative stress-dependent activation of the Src kinase has been causally linked to Akt activation and subsequent autophagy flux impairment, in the mdx mouse[13]. Immunoblot analysis revealed that JQ1 treatment reduced Src phosphorylation in TAs from mdx mice (Fig. 3e), suggesting that autophagy rescue may be related to an upstream decrease in oxidative stress. Taken together these data suggest that JQ1 treatment restores autophagy in the mdx skeletal muscles.

**JQ1 restrains oxidative stress.** To further investigate JQ1 impact on ROS metabolism in the mdx muscle, we asked whether ROS levels were affected by JQ1 treatment in C2C12 cells, in which previously published RNA-seq datasets show that BRD2/3/4 are highly expressed, with BRD2 transcript being the most abundant followed by BRD4 and then BRD3[23,44–46]. We first performed immunofluorescence experiments with anti-8-OHdG antibody, which confirmed increased oxidative stress in the mdx muscle, when compared to control muscles[47]. JQ1 treatment significantly reduced 8-OHdG immunoreactivity, in TAs (Fig. 4a).

To investigate whether JQ1 was able to protect cells from a second source of ROS, we also employed an in vitro model in which oxidative stress was induced by hydrogen peroxide ($H_2O_2$) treatment in C2C12 cells. In agreement with our in vivo findings, JQ1 treatment prevented $H_2O_2$-induced oxidative stress, as revealed by 8-OHdG staining (Fig. 4b). Moreover, $H_2O_2$ treatment was able to impair autophagy in C2C12 myotubes in which the autophagic flux was blocked by chloroquine, as indicated by the pattern of LC3II and p62. Nevertheless, co-administration of JQ1 to $H_2O_2$-treated cells restored LC3II abundance and reduced p62 levels, as observed for mdx muscles in Fig. 4c.

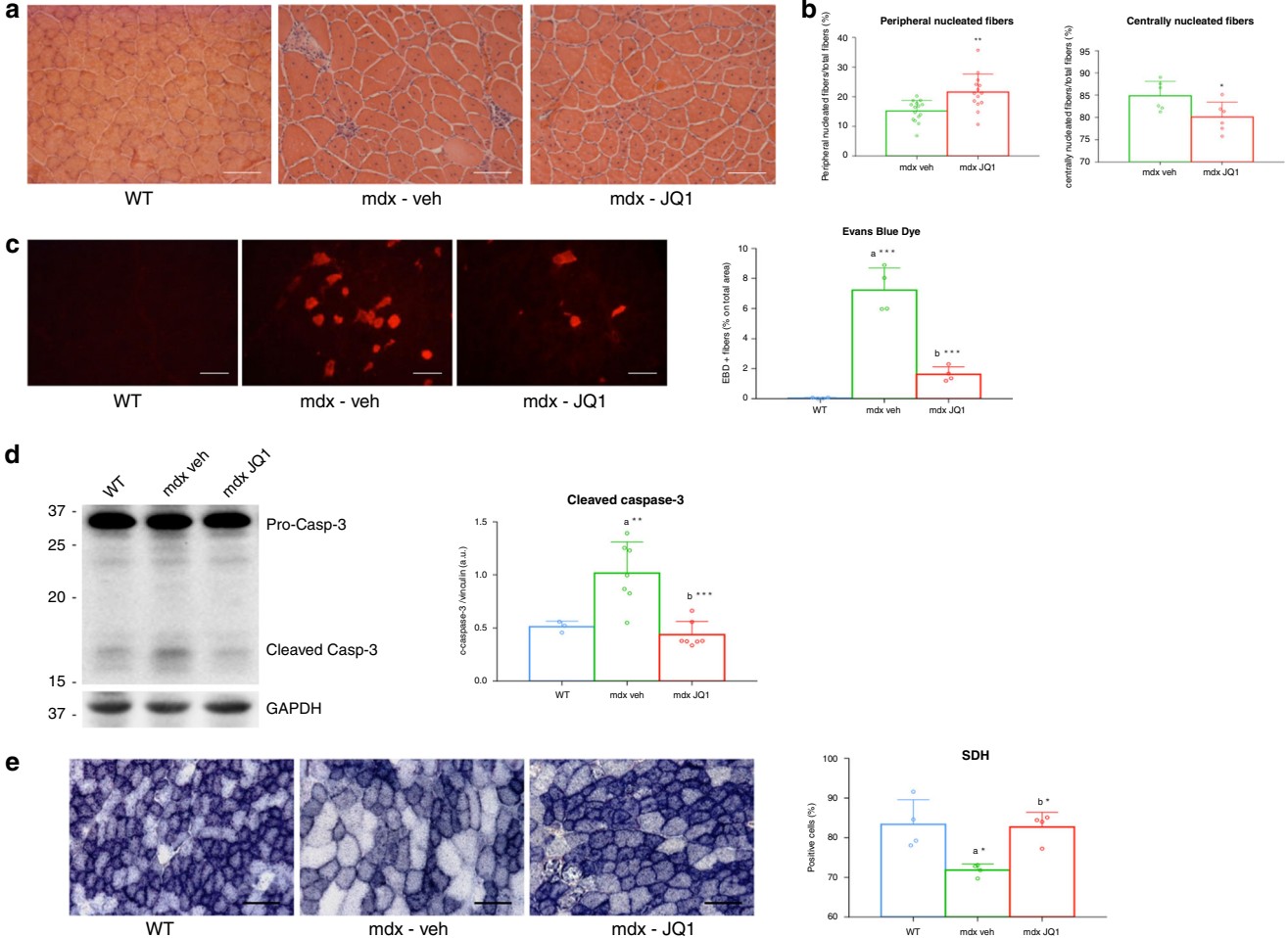

**Fig. 2 JQ1 treatment leads to morphological improvements in mdx mice. a** Hematoxylin/Eosin staining of TA muscles from JQ1- and vehicle-treated mdx mice. JQ1 was chronically administered for two weeks. Scale bar: 50 μm. **b** JQ1 treatment increases the number of intact fibers in mdx TAs. Data are expressed as the mean ± SD, $n = 3$ animals, *$P < 0.05$ and **$P < 0.01$ was determined by using unpaired two-sided $t$-test. Total fibers and percentage of peripheral ($n = 15$ sections examined from $n = 3$ animals for each experimental group) and centrally nucleated ($n = 6$ sections examined from $n = 3$ animals for each experimental group) fibers per area were evaluated from Hematoxylin/Eosin staining of TA muscles ($n = 3$ for each experimental group). **c** JQ1 treatment strongly decreases the number of damaged myofibers, as observed by the decreased uptake of Evans blue dye from TA mdx muscles ($n = 4$ for each experimental group). Scale bar: 50 μm. ***denotes $P < 0.001$ and was determined by 1 way-Anova followed by Tukey's post hoc test. **d** Immunoblot showing pro-Caspase-3 and cleaved Caspase-3 in TAs from control, vehicle-, and JQ1-treated mice. Right panel: quantification of band intensity was performed with ImageJ. Data are expressed as the mean ± SD, $n = 7$ animals for mdx groups, $n = 3$ animals for WT. **$P < 0.01$ and ***$P < 0.001$ were determined by using one-way ANOVA followed by Tukey's post hoc test. a indicates statistical significance compared to Control group; b indicates statistical significance compared to the mdx mice animal group. **e** SDH staining from muscle deep region of TAs, from control, vehicle- and JQ1-treated mice. Scale bar: 50 μm. Right panel: quantification of SDH staining intensity. Data are expressed as the mean ± SD, $n = 4$ animals per group. a and b as defined in (d). *Denotes $P < 0.05$ and was determined by 1 way-Anova followed by Tukey's post hoc test.

In several cultured cell lines, defined doses of $H_2O_2$ were able to induce a reduction in p-AMPK levels[48–51]. Similarly, in C2C12 myotubes, oxidative stress generated by $H_2O_2$ administration led to a decrease in AMPK and AMPK-dependent Ulk1 phosphorylation, and was associated with an increase in Akt phosphorylation (Ser473); however, all these events were prevented by JQ1 co-treatment (Fig. 4d).

Oxidative stress was previously shown to induce Sirt1 carbonylation and proteasomal degradation[52]. $H_2O_2$ treatment reduced Sirt1 abundance in C2C12 myotubes, as reported[53] (Fig. 4d). However, JQ1 co-treatment prevented $H_2O_2$-induced Sirt1 protein reduction, as revealed by a degradation assay (Fig. 4d, Supplementary Fig. 4A). Moreover, to link Sirt1 activity with p-AMPK levels, we challenged Sirt1 function with nicotinamide (NAM) and we observed that JQ1-mediated recovery in AMPK phosphorylation was dependent on Sirt1

activity, in $H_2O_2$-treated myotubes. Coherently, AMPK-dependent phosphorylation of Ulk1 was also influenced by Sirt1 inhibition (Supplementary Fig. 4B). To test whether JQ1 is effective when oxidative stress is already established, we administered JQ1 after treating C2C12 myotubes with $H_2O_2$ for 2 hours, a sufficient time to induce oxidative stress in C2C12 cells (Supplementary Fig. 5A). The modulation of Sirt1, p62, LC3 as well as of phosphorylated AKT, AMPK and Ulk1 were similar to the one obtained when cells were pretreated with JQ1, followed by $H_2O_2$ stimulation (Supplementary Fig. 5B, C). In addition, JQ1 treatment alone did not affect p-AKT and p62 levels, but it increased Sirt1, lipidated LC3 levels, AMPK and Ulk1 phosphorylation (Supplementary Fig. 5B–D). In this experimental setting, we confirmed that NAM treatment prevented AMPK activation and Ulk1 phosphorylation (Supplementary Fig. 5C). Moreover, Sirt1 pharmacological blockade hindered LC3 accumulation and

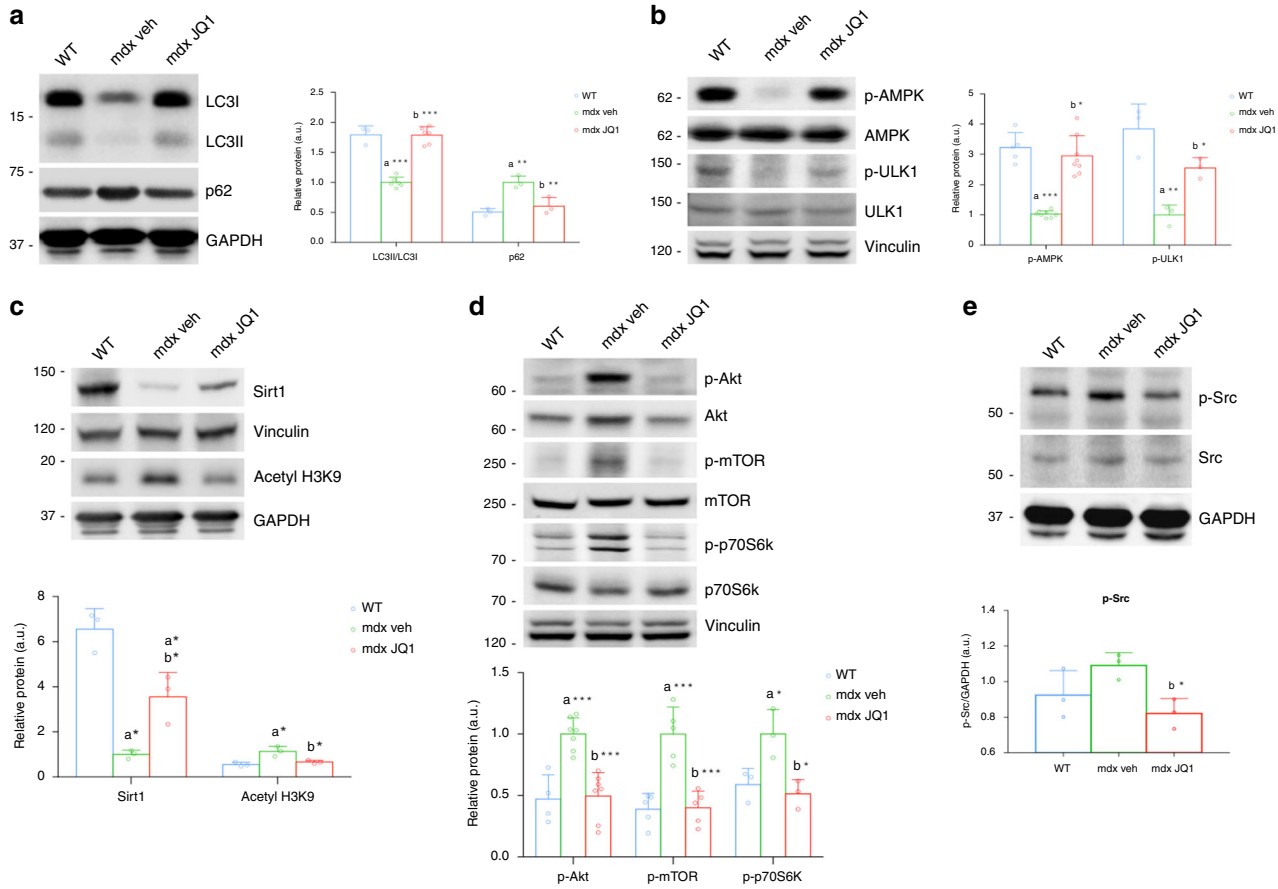

**Fig. 3 BET blockade rescues autophagy in the mdx muscle. a** Representative western blot for LC3 (WT, $n = 3$; mdx, $n = 6$; mdx+JQ1, $n = 6$) and p62 ($n = 3$ for each experimental group) in TA extracts of control and vehicle-and JQ1-treated mice Lower panel: quantification of normalized band intensity. Data represent means ± SD. GAPDH serves as a loading control. **b** Representative western blot for AMPK and p-AMPK (WT, $n = 5$; mdx, $n = 8$; mdx+JQ1, $n = 8$), Ulk1 and p-Ulk1 (Ser555) ($n = 3$ for each experimental group) in TA extracts of control and vehicle-and JQ1-treated mice. Lower panel: quantification of normalized band intensity. Data represent means ± SD. Vinculin serves as a loading control. **c** Representative western blot for Sirt1 and H3K9Ac in TA extracts of control and vehicle-and JQ1-treated mice. GAPDH and Vinculin serve as loading controls. Lower panel: quantification of normalized band intensity. Data represent means ± SD. $n = 3$ animals per group. **d** Representative western blot for Akt and p-Akt (Ser473) (WT, $n = 4$; mdx, $n = 7$; mdx+JQ1, $n = 7$), mTOR and p-mTOR (Ser2448) ($n = 5$ for each experimental group), p70S6K and p-70SK6 (Thr389) ($n = 3$ for each experimental group) in TA extracts of control and vehicle-and JQ1-treated mice Right panel: quantification of normalized band intensity. Data represent means ± SD. Vinculin serves as a loading control. **e** Representative western blot for Src and p-Src in TA extracts of control and vehicle-and JQ1-treated mice Lower panel: quantification of normalized band intensity. GAPDH serves as a loading control. Data represent means ± SD. $n = 3$ per experimental group. In all panels, statistical analysis was performed by using one-way ANOVA followed by Tukey's post hoc test. In all relevant panels, *$P < 0.05$; **$P < 0.01$; ***$P < 0.001$. a indicates statistical significance compared to control; b indicates statistical significance compared to mdx.

p62 downregulation in cells in which oxidative stress was induced by $H_2O_2$ followed by JQ1 treatment (Supplementary Fig. 5D), further supporting the idea that Sirt1 plays a key role in AMPK activation and autophagy regulation.

Collectively, these data suggest that JQ1 treatment prevented and tempered perturbation in ROS metabolism both in the dystrophic muscle and in an in vitro model of oxidative stress, thus restoring Sirt1 levels, AMPK and Akt phosphorylation state, and autophagy.

**BETs pharmacological blockade prevents NADPH subunit transcription upregulation.** NADPH oxidase subunits are over-expressed in the adult mdx muscle and they represent a major source of ROS production in the mdx muscle[7,54,55]. Accordingly, we observed that mRNA levels of the Nox2, Nox4, p67-phox, and p47-phox subunits were higher in TAs from mdx mice, when compared to those of control mice. JQ1 treatment in mdx mice restored the transcripts of NADPH oxidase subunits to the level of control mice (Fig. 5a). Similarly, Nox2 and p67-phox protein

levels were higher in TAs from mdx mice when compared to control mice, but their levels decreased in muscles from JQ1-treated mdx mice (Fig. 5b and Supplementary Fig. 6A). Likewise, when we treated isolated mdx myofibers with JQ1, we observed a downregulation in transcript levels of the NADPH oxidase subunits, suggesting that JQ1-mediated transcriptional regulation can occur in myofibers, once depleted of infiltrating mononuclear cells (Fig. 5c).

We also analyzed transcripts of the NADPH oxidase subunits in immortalized myoblasts from young DMD donors and observed that their levels significantly decreased following JQ1 treatment (Fig. 5d). Likewise, interrogation of RNA-seq datasets, revealed that Nox2, p47-phox, and p67-phox transcript levels increased in skeletal muscles from DMD patients[36], extending the relevance of our findings to the human pathology (Fig. 5e).

In the mdx muscle, JQ1 administration did not affect mRNA and protein levels of Nrf2, a transcription factor that plays a key role in the antioxidant response pathway (Supplementary Fig. 7A, B). Likewise, JQ1 treatment did not alter the transcriptional

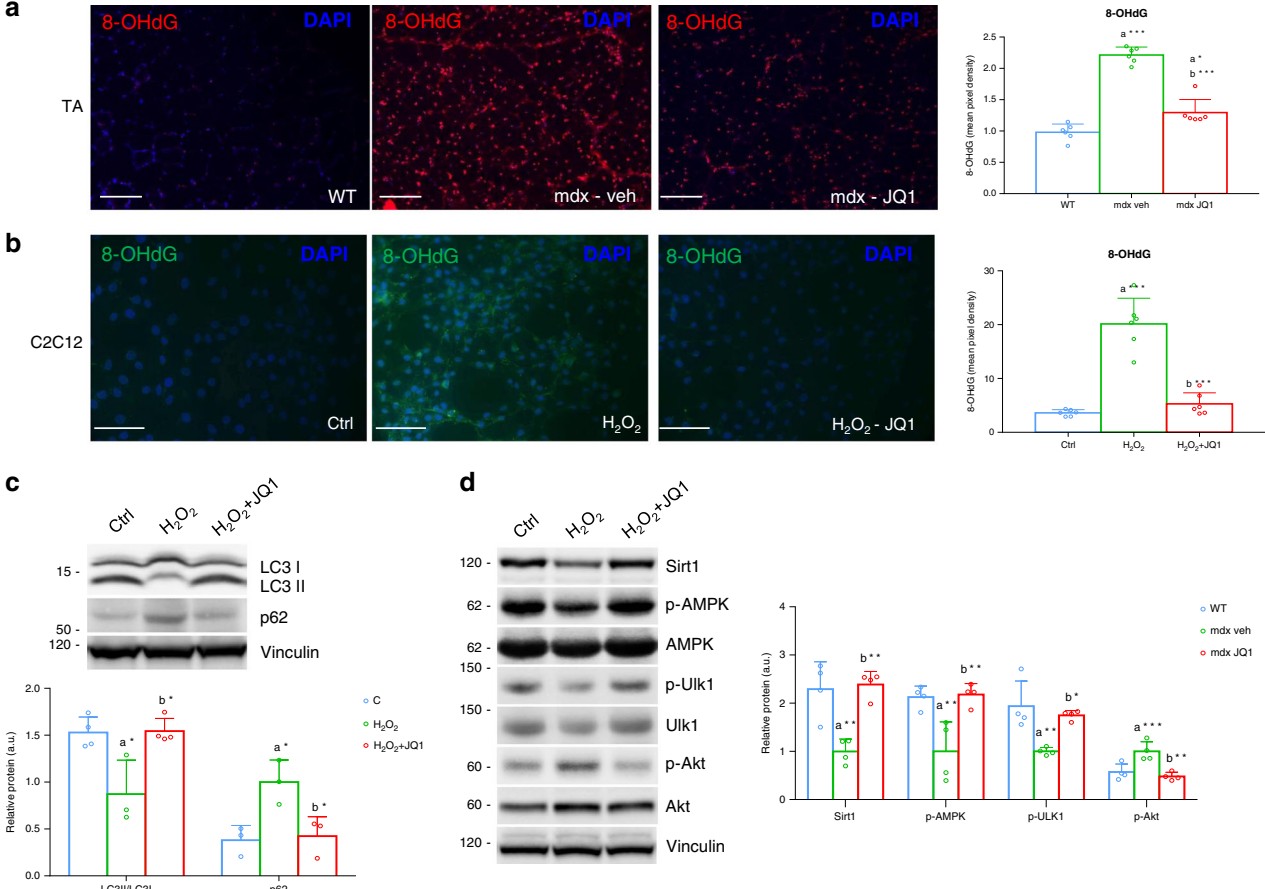

**Fig. 4 JQ1 decreases oxidative stress in the mdx muscle and in $H_2O_2$-treated cells. a** Representative images of 8-OHdG staining of muscle cross-sections in control, vehicle- and JQ1-treated TAs ($n = 6$ sections examined from $n = 3$ animals for each experimental group). Scale bar: 50 μm. Right panel: quantification of the staining. Data are expressed as a mean ± SD. a indicates statistical significance compared to Control group; b indicates statistical significance compared to the mdx mice animal group. **b** Representative images of 8-OHdG staining of C2C12 cells. Cells were pretreated with JQ1 (200 nM) for 24 h and then stimulated with $H_2O_2$ for 24 h. Scale bar: 50 μm. Right panel: quantification of the staining derived from three independent experiments. $n = 6$ fields examined from $n = 3$ independent experiments. Data are expressed as a mean ± SD. a indicates statistical significance compared to Control cells; b indicates statistical significance compared to the $H_2O_2$-treated C2C12. **c** Representative western blot for LC3 (**c**, $n = 4$; $H_2O_2$, $n = 3$; $H_2O_2 + JQ1$, $n = 4$) and p62 ($n = 3$ independent experiments) in C2C12 myotube extracts of control and $H_2O_2$- and $H_2O_2$/JQ1-treated cells for 24 h. Myotubes were pretreated with JQ1 (200 nM) for 24 h and then stimulated with $H_2O_2$ for 24 h. In order to study the autophagy flux, the experiment was performed by pre-treating cells with 30 μM of the lysosomotropic agent chloroquine. Vinculin serves as a loading control. Right panel: quantification of normalized band intensity derived from three different experiments. Data represent means ± SD. a indicates statistical significance compared to Control cells; b indicates statistical significance compared to the $H_2O_2$-treated C2C12. **d** Representative western blot for Sirt1, p-AMPK, AMPK, Ulk1, p-Ulk1 (Ser555), Akt, p-Akt (Ser473) in C2C12 myoblast extracts of control and $H_2O_2$- and $H_2O_2$/JQ1-treated cells. Myotubes were pretreated with JQ1 (200 nM) for 24 h and then stimulated with $H_2O_2$ for 24 h. Vinculin serves as a loading control. Right panel: quantification of normalized band intensity derived from at least four different experiments. Data represent means ± SD, $n = 4$ independent experiments. a indicates statistical significance compared to Control cells; b indicates statistical significance compared to the $H_2O_2$-treated C2C12. In all panels, statistical analysis was assessed by using one-way ANOVA followed by Tukey's post hoc test. *$P < 0.05$; **$P < 0.01$; ***$P < 0.001$.

regulation of Nrf2 targets, Hmox1, Gclm and Gclc (Supplementary Fig. 7C), suggesting that restoration of ROS metabolism was not ascribed to the transcriptional activation of anti-oxidant genes.

Next, we mimicked ROS induction in C2C12 myotubes, by treating the cells with 250 μM $H_2O_2$ for 24 h. $H_2O_2$ is a pro-oxidant that contributes to the generation of a vicious cycle of ROS production by upregulating NADPH oxidase subunits through the activation of redox-sensitive transcription factors[50,56]. $H_2O_2$ treatment promoted p67-phox, p47-phox, Nox2 and Nox4 transcription in C2C12 myotubes, whereas JQ1 co-treatment prevented their transcriptional activation (Fig. 5f). $H_2O_2$-induced upregulation of Nox2 and p67-phox proteins was also detectable and it was abrogated by JQ1 pre-treatment (Fig. 5g and Supplementary Fig. 6B). Similarly, NADPH oxidase subunits

transcriptional upregulation was hindered when JQ1 administration followed $H_2O_2$ treatment (Supplementary Fig. 8A).

JQ1 binds the BET proteins (BRD2, BRD3, BRD4, BRDT) with different affinity and it is potentially capable of displacing BRD2/BRD3/BRD4 from chromatin[57]. To define which BET protein plays a major role in NADPH oxidase subunits modulation, we employed a siRNA approach in C2C12 myoblasts. BRD2 knockdown reduced the transcript levels of Nox2, Nox4, p47-phox, and p67-phox, while BRD4 depletion did not affect Nox4 mRNA but decreased Nox2, p47-phox, and p67-phox expression. BRD3 did not influence NADPH subunits expression levels (Fig. 5h and Supplementary Fig. 9A, B). Because of BRD2 and BRD4 ability to modulate NADPH oxidase subunits, we performed ChIP assays for these two BET proteins in skeletal

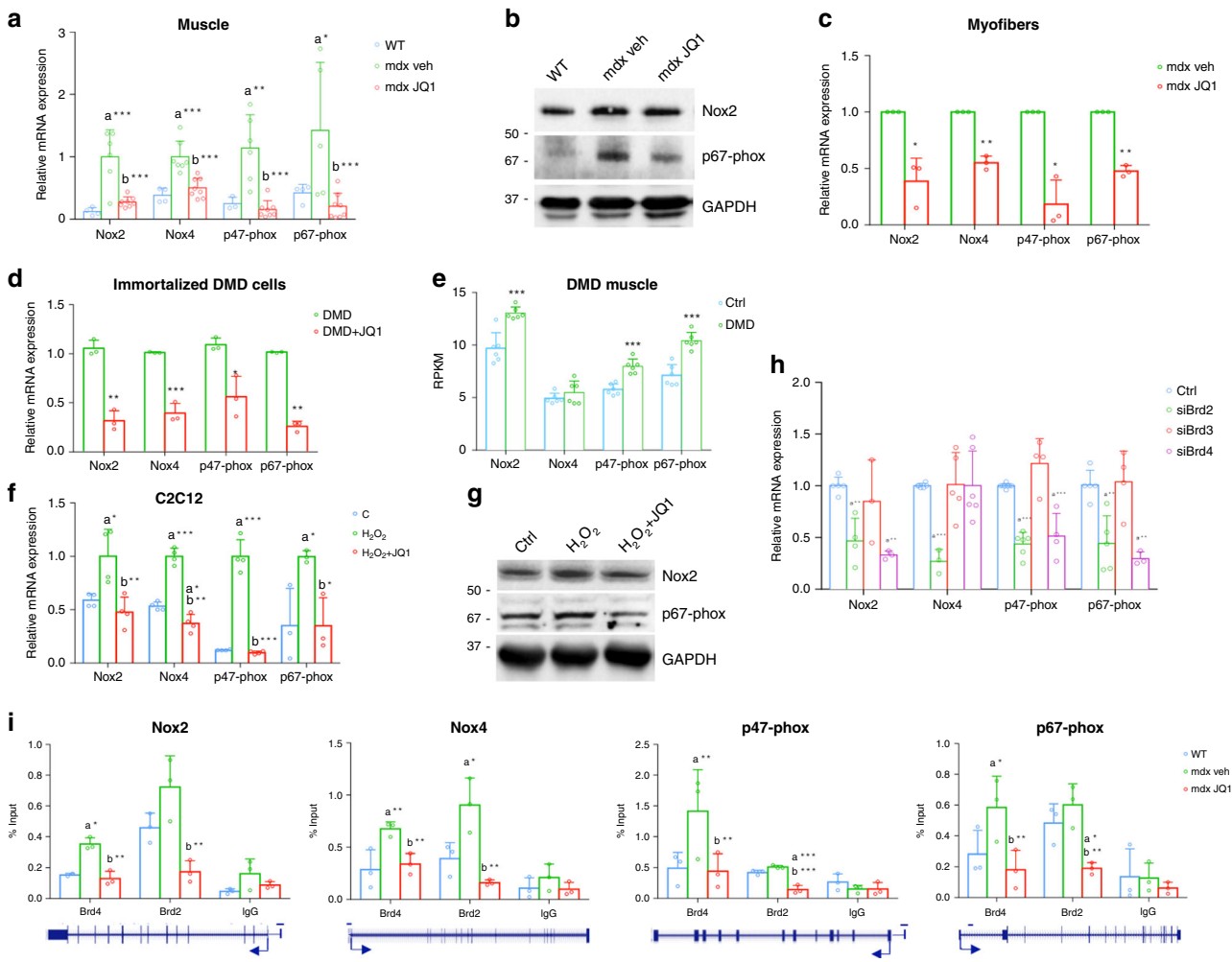

**Fig. 5 JQ1 reduces the transcriptional upregulation of NADPH oxidase subunits. a** qRT-PCR analysis of Nox2 (WT, $n = 4$; mdx, $n = 6$; mdx+JQ1, $n = 8$), Nox4 (WT, $n = 4$; mdx, $n = 7$; mdx+JQ1, $n = 8$), p47-phox (WT, $n = 3$; mdx, $n = 6$; mdx+JQ1, $n = 8$) and p67-phox (WT, $n = 4$; mdx, $n = 5$; mdx+JQ1, $n = 8$) mRNAs in TAs from control, vehicle-, and JQ1-treated mice. Data are normalized against HPRT and expressed as the mean ± SD. **b** Representative images of immunoblot analysis for Nox2 and p67-phox in TAs from control, vehicle-, JQ1-treated mice. GAPDH serves as a loading control. WT animals: $n = 3$ for each experimental group. **c** qRT-PCR analysis of NADPH oxidase subunit mRNAs in myofibers isolated from mdx EDL muscles and treated with 200 nM JQ1 for 16 h. Data are normalized against HPRT and expressed as the mean ± SD, $n = 3$ animals. **d** qRT-PCR analysis of NADPH oxidase subunit mRNAs in immortalized DMD myoblast cells treated with 200 nMJQ1 for 24 h. Data are normalized against GAPDH and expressed as the mean ± SD, $n = 3$ immortalized cell lines. **e** RPKM expression levels of NADPH subunit transcripts in published RNA-Seq dataset for DMD ($n = 6$) and healthy donors ($n = 6$). Data are expressed as a mean ± SD. **f** qRT-PCR analysis of Nox2, Nox4, p47 phox ($n = 4$ for each experimental group) and p67-phox ($n = 3$ for each experimental group) mRNAs in C2C12 myotubes cells were pretreated with 200 nM JQ1 and then co-treated with 250 µM $H_2O_2$ for 8 h. Data are normalized against GAPDH and expressed as the mean ± SD. **g** Representative images of immunoblot analysis for Nox2, p67-phox and BRD4 in C2C12 myoblasts treated as in (**e**). GAPDH serves as a loading control. **h** qRT-PCR analysis of Nox2 (Ctrl, $n = 5$; siBrd2, $n = 4$; siBrd3, $n = 3$; siBrd4, $n = 3$), Nox4 (Ctrl, $n = 5$; siBrd2, $n = 3$; siBrd3, $n = 5$; siBrd4, $n = 5$), p47-phox (Ctrl, $n = 5$; siBrd2, $n = 6$; siBrd3, $n = 4$; siBrd4, $n = 4$) and p67-phox (Ctrl, $n = 5$; siBrd2, $n = 5$; siBrd3, $n = 4$; siBrd4, $n = 3$) subunit mRNAs in C2C12 cells in which BRD2, BRD3, BRD4 levels were decreased by siRNAs transfection. Data are normalized against GAPDH and expressed as the mean ± SD. **i** ChIP assay with BRD2 and BRD4 antibodies in control, vehicle- and JQ1-treated muscles showing recruitment at regulatory regions of Nox2, Nox4, p47-phox and p67-phox genes. Data represent mean ± SD, $n = 3$ animals A schematic representation below the diagrams shows the region amplified in ChIP. In all panels, statistical significance was determined by using one-way ANOVA followed by Tukey's post hoc test. a indicates statistical significance compared to the group presented in the first column; b indicates statistical significance compared to the group presented in the second column. $*P < 0.05$; $**P < 0.01$; $***P < 0.001$.

muscles of control, vehicle- and JQ1-treated mdx mice. We amplified chromatin regulatory regions that we previously found to be occupied by BRD4 in TA of control mice by ChIP-seq assays[32], which were shown to include previously described regulatory regions[58–63]. These experiments disclosed that BRD2 occupies regulatory regions of Nox2, p47-phox, and p67-phox at a comparable level in control and mdx mice, while BRD2 occupancy at the Nox4 promoter and BRD4 recruitment at the regulatory regions of Nox2, Nox4, p47-phox, p67-phox subunits

significantly increased in mdx skeletal muscles. In JQ1-treated mdx mice, BRD4 and BRD2 occupancy was lost in all loci analyzed (Fig. 5i). Overall, these findings revealed that BRD4 and BRD2 occupy the NADPH oxidase subunits regulatory regions in the mdx muscle and modulate their transcription. Conversely, BETs pharmacological blockade prevents NADPH oxidase subunits transcriptional upregulation in the mdx muscles, in isolated mdx myofibers, in an in vitro model of oxidative stress, and in DMD immortalized myoblasts.

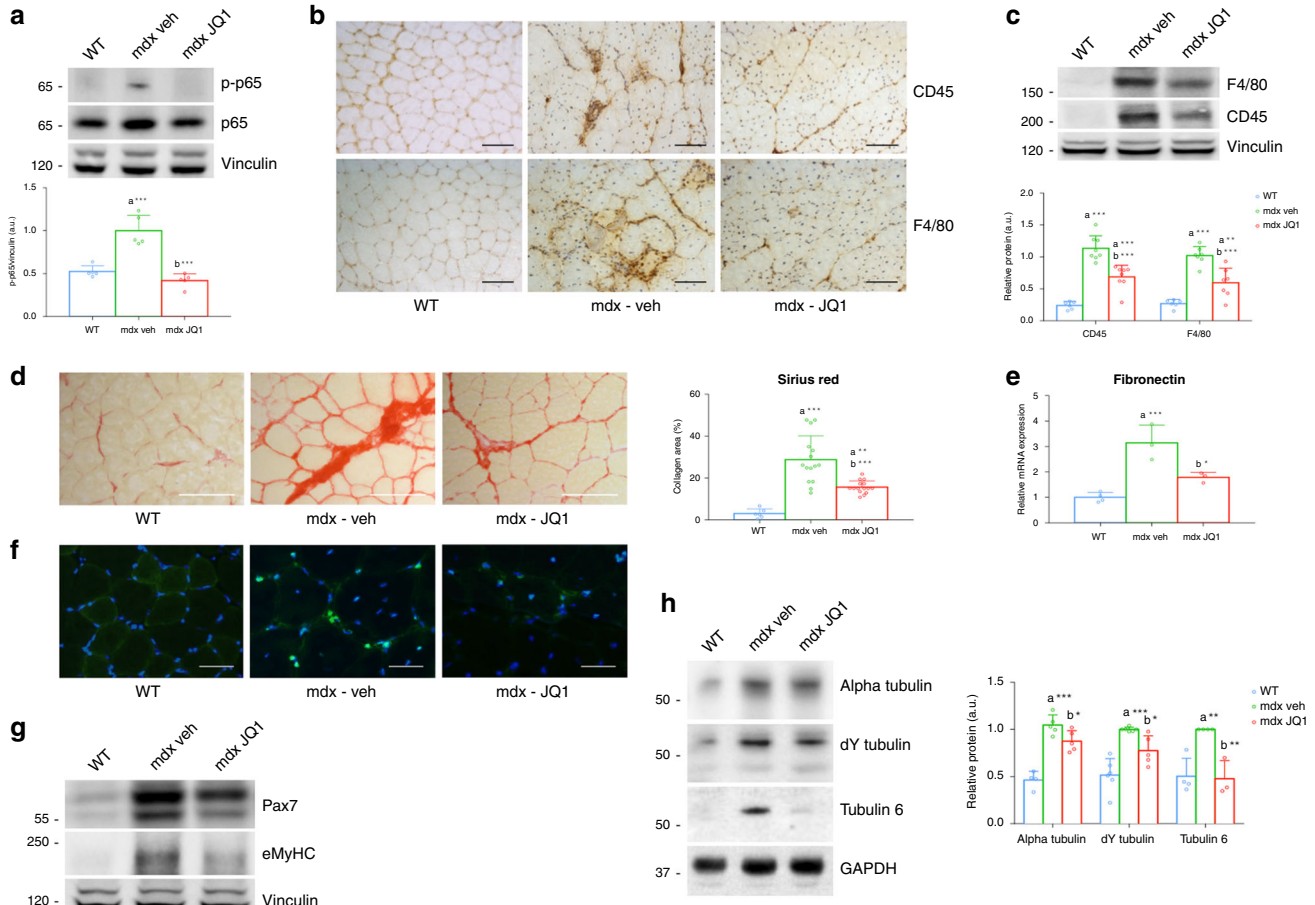

**Fig. 6 JQ1 treatment ameliorates pathological phenotypes in dystrophic muscle. a** JQ1 administration decreases NFkB p65-Ser536 phosphorylation in TA muscles from mdx mice ($n = 5$ for each experimental group). Lower panel: quantification of signals was performed with ImageJ. Data are expressed as the mean ± SD. ***$P < 0.001$. Statistical significance was determined by using one-way ANOVA followed by Tukey's post hoc test. **b** Immunohistochemical evaluation of F4/80 and CD45 ($n = 3$ for each experimental group). Scale bar: 50 μm. **c** Immunoblot analysis of F4/80 (WT, $n = 6$; mdx, $n = 7$; mdx+JQ1, $n = 7$) and CD45 (WT, $n = 6$; mdx, $n = 8$; mdx+JQ1, $n = 8$) in muscles from control, vehicle- and mdx-treated mice. Vinculin serves as a loading control. Lower panel: quantification of signals was performed with ImageJ. Data are expressed as the mean ± SD. **$P < 0.01$, ***$P < 0.001$. Statistical significance was determined by using one-way ANOVA followed by Tukey's post hoc test. **d** Sirius red staining shows attenuation of fibrosis in JQ1-treated mice ($n = 2$ sections examined from $n = 3$ WT animals; mdx mice, $n = 5$ sections examined from $n = 3$ mdx-veh and mdx-JQ1 animals). Scale bar: 50 μm. Right panel: quantification of staining. **$P < 0.01$, ***$P < 0.001$. Statistical significance was determined by using one-way ANOVA followed by Tukey's post hoc test. **e** qRT-PCR analysis of Fibronectin mRNA in TA muscles from control ($n = 4$), vehicle- ($n = 3$) and mdx-treated ($n = 3$) mice. Data are normalized against HPRT and expressed as the mean ± SD. *$P < 0.05$, ***$P < 0.001$. Statistical significance was determined by using one-way ANOVA followed by Tukey's post hoc test. **f** Immunofluorescence analysis with an antibody raised against the regeneration marker Pax7 in TA cross-sectional section of WT and vehicle- or JQ1-treated mdx mice. $n = 3$ animals for each experimental group. Scale bar: 25 μm. **g** Immunoblot analysis of Pax7 (WT, $n = 7$; mdx, $n = 6$; mdx+JQ1, $n = 7$) and eMyHC (WT, $n = 3$; mdx, $n = 3$; mdx+JQ1, $n = 3$) levels in TA of WT, vehicle- and JQ1-treated mdx mice. **h** Immunoblot analysis of alpha tubulin (WT, $n = 4$; mdx, $n = 5$; mdx+JQ1, $n = 5$), dY tubulin (WT, $n = 6$; mdx, $n = 5$; mdx+JQ1, $n = 5$) and tubulin6 (WT, $n = 4$; mdx, $n = 4$; mdx+JQ1, $n = 3$) levels in TA of WT, vehicle- and JQ1-treated mdx mice. Right panel: quantification of signals was performed with ImageJ. Data represent mean ± SD. *$P < 0.05$, **$P < 0.01$, ***$P < 0.001$. Statistical significance was determined by using one-way ANOVA followed by Tukey's post hoc test.

**JQ1 treatment improves muscle physiopathology in the mdx muscle.** We further tested whether JQ1 treatment was accompanied by an overall improvement of the pathological abnormalities of the mdx skeletal muscle. In mdx mice, the p65 subunit of NFkB is activated by phosphorylation at Ser536 (Fig. 6a), as a consequence of increased intracellular Ca$^{2+}$[64] and altered ROS production[64]. NFkB activation, in turn, promotes the transcription of pro-inflammatory cytokines. Remarkably, 2 weeks of JQ1 treatment resulted in a significant reduction in p65 phosphorylation (Fig. 6a). Inflammation was mitigated by JQ1 treatment, as shown by immunostainings for the leukocyte antigen CD45 and the macrophage marker F4/80, which signals were attenuated following BET blockade (Fig. 6b). These findings were supported by the JQ1-dependent decrease in F4/80 and CD45 protein levels

in TAs of mdx mice (Fig. 6c). The reduction of inflammatory infiltrate in TA muscles of JQ1-treated animals was coupled to the suppression of TNFα (Supplementary Fig. 10A) and of IL6 pathways (Supplementary Fig. 10B–D), two crucial pro-inflammatory signaling events involved in the pathogenesis of DMD. Reduction in Fibronectin transcript levels and decreased collagen fibers deposition in TAs, revealed by Sirius red staining, indicated that reduced inflammation was accompanied by a decrease in fibrosis following JQ1 administration (Fig. 6d, e). To evaluate the impact of JQ1 treatment in older animals, we daily treated 11 months old mdx mice with JQ1 (20 mg/day) by intraperitoneal injection, for 4 weeks. At this stage of the disease progression, JQ1 administration led to a reduction in the transcript levels of inflammatory markers TNFα and IL6

(Supplementary Fig. 11A), which was paralleled by a decrease in the levels of CD45 and F4/80 proteins (Supplementary Fig. 11B), as well as of inflammatory infiltrate (Supplementary Fig. 11C). BET blockade led to a trend towards increasing the number of peripheral nucleated fibers and reducing the centrally nucleated fibers, although not significantly (Supplementary Fig. 11D). Fibrosis was reduced in 12-month-old JQ1-treated mdx TAs, as shown by Sirius red staining (Supplementary Fig. 11E). Transcript levels of NADPH oxidase subunits and collagen 1α1 were also reduced following JQ1 administration (Supplementary Fig. 11F, G). Overall, these results show that, in the mdx mouse model, JQ1 treatment has a beneficial impact also when the disease phenotype is aggravated.

Furthermore, in 12-week-old mdx mice, reduced muscle damage and inflammation correlated with a more modest increase in markers of regeneration (eMyHC, Pax7, MyoD, and Myogenin) in TAs of JQ1 versus vehicle-treated mdx mice, as observed by protein (Fig. 6f, g and Supplementary Fig. 12A, C) and RNA levels (Supplementary Fig. 12B). These data are in agreement with the observed decrease in centrally nucleated fibers (Fig. 2b). Since a reduction in central nucleated fibers is a hallmark of improved muscle histology in the dystrophic muscle, our data suggest that reduced muscle damage was accompanied by decreased regeneration. In vitro, JQ1 (200 nM) treatment of satellite cells did not prevent their ability to differentiate, nor significantly decreased their proliferation rate (Supplementary Fig. 12D, E).

In DMD muscles, dystrophin absence alters the cytoskeleton, which results as a disorganized net of denser microtubules. Since the microtubules network conveys mechanotransduction signals to Nox2-dependent enhancement of ROS[17,36,65–67] in adult mdx muscles, we asked whether JQ1 treatment was able to correct microtubules anomalies that contribute to the DMD pathology. We confirmed that total and de-tyrosinated alpha-tubulin is increased in adult mdx muscles, and we found that JQ1 treatment decreased both alpha-tubulin and de-tyrosinated tubulin (Fig. 6h). Tubulin6 protein significantly increased in adult mdx TAs when compared to control animals[66,67], and JQ1 reduced its levels to the ones of control mice (Fig. 6h).

**Amelioration of muscle functional performance following JQ1 treatment in mdx mice.** In agreement with the overall reduced muscle damage, JQ1-treated mdx mice significantly increased resistance to fatigue in the treadmill test, and they showed a substantial amelioration in endurance. In addition, overall evaluation of muscle force employing the inverted screen and the wire tests also showed improvements in muscle performance and in vivo force, as early as 2 and 1 week of treatment, respectively (Fig. 7a). Significant amelioration was maintained for one week in the treadmill, wire and inverted screen tests after JQ1 treatment withdrawal and overall motor function of JQ1-treated mdx mice returned to levels comparable to vehicle mdx-mice only three weeks after JQ1 withdrawal (Fig. 7b).

Taken together, our findings demonstrate that BET inhibition ameliorates the physiopathological defects of dystrophic skeletal muscle, suggesting that BET targeting may be beneficial for patients with muscular dystrophies.

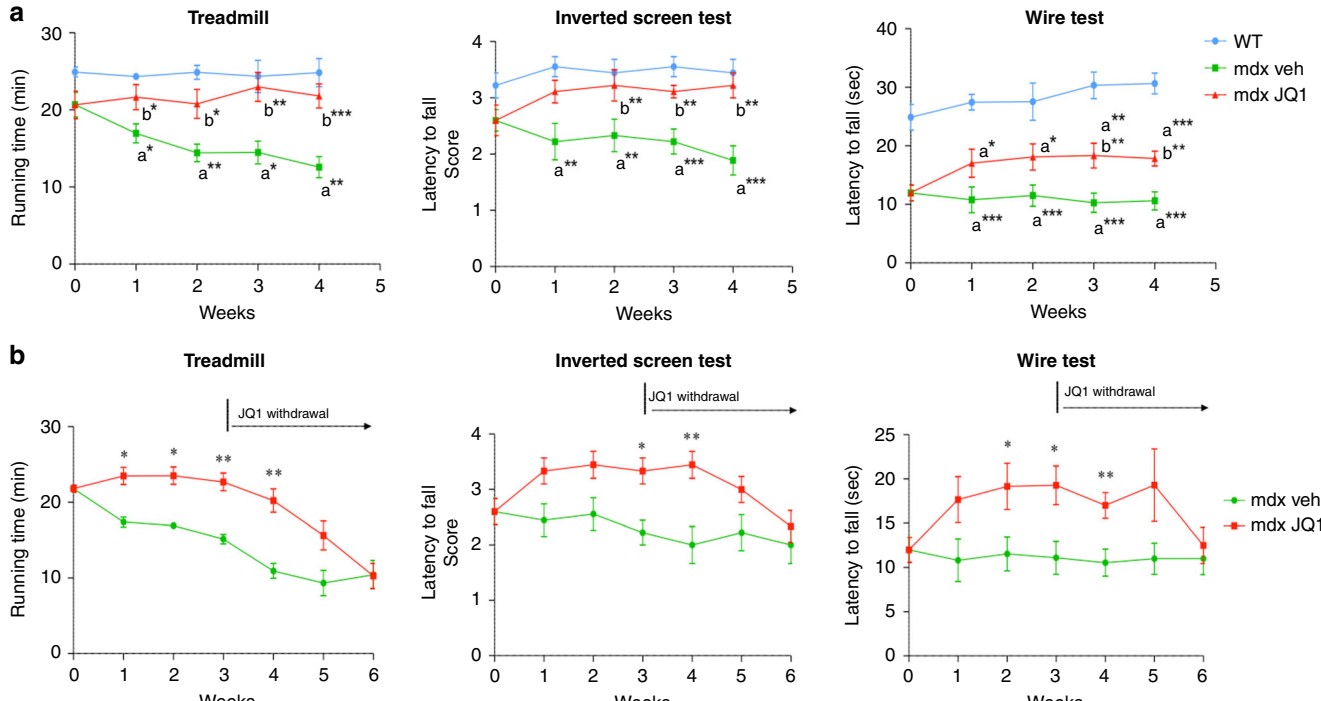

**Fig. 7 Functional amelioration mediated by JQ1 persists following withdrawal. a** Treadmill (WT: $n = 3$; mdx veh: $n \geq 7$; mdx JQ1: $n \geq 8$), inverted screen ($n = 9$ for each experimental group) and wire (wt animals: $n = 9$; mdx mice: $n = 18$ for each group) tests were performed on control, vehicle- and mdx-treated mice. Data are expressed as the mean ± SEM. *$P < 0.05$, **$P < 0.01$ and ***$P < 0.001$ were determined by one-way ANOVA followed by Tukey's post hoc test for the treadmill test and with Kruskal–Wallis test followed by Dunns post hoc for wire and inverted screen tests. a indicates statistical significance compared to Control group; b indicates statistical significance compared to the mdx mice animal group. **b** Mice were treated with JQ1 or vehicle for 3 weeks and treadmill ($n = 3$ for each experimental group), inverted screen ($n = 9$ for each group) and wire ($n \geq 10$ for each group) tests were performed once a week, and for additional 3 weeks after JQ1 withdrawal. Data are expressed as the mean ± SEM. *$P < 0.05$ and **$P < 0.01$ indicate statistical significance versus mdx-vehicle group, and were determined by one-way ANOVA followed by Tukey's post hoc test.

## Discussion

Alteration in ROS metabolism has been identified as an early event in Duchenne muscular dystrophy, leading to myonecrosis, muscle damage and inflammatory cell infiltration[7]. Increased oxidative stress results from an unbalance between increased production of reactive oxygen/nitrogen species and an insufficient antioxidant response, leading to myofiber damage and tissue degeneration. Muscle biopsies from DMD patients show increased oxidative stress compared to controls[14,15]; in mdx muscles, increased NADPH oxidase Nox2 activity and Src kinase activation cause an increase in oxidative stress[7,16]. Here we show that BRD4 is involved in transcriptional activation of different subunits of the NADPH oxidase complex in the mdx muscle and that BETs pharmacological inhibition dramatically reduces oxidative stress and ameliorates skeletal muscle homeostasis and muscle function. Increased Nox2 activity and oxidative stress has been recently causally linked to autophagy impairment in the mdx muscle. Furthermore, Nox2 genetic ablation ameliorates pathological and functional phenotypes in dystrophic muscle[13]. Accordingly, we show that JQ1 administration rescues autophagy and restricts muscle damage, preventing muscle inflammation and fibrosis (Fig. 8). In Duchenne muscular dystrophy several processes are deregulated and concur to exacerbate the dystrophic phenotype. Moreover, phagocytic inflammatory cells are a significant source of ROS and the reciprocal stimulation between oxidative stress and inflammation rapidly amplifies the axis leading to muscle degeneration, throughout disease progression. In this scenario, the transcription factor p65 plays a pivotal role, since it is a redox-sensing transcription factor, activated by ROS increase[68] and it concurrently regulates activation of inflammatory transcripts. BRD4 has been shown to play a key role as a cofactor in promoting transcriptional activation of inflammatory genes, in sepsis as well as atherogenesis[33–35]. Therefore, BRD4 may not only regulate p65-regulated genes indirectly through p65 ROS-mediated activation, but also directly tempering p65 mediated transcriptional activation at certain target genes such as inflammatory cytokines. BET inhibitors ability to concurrently counteract different processes, such as oxidative stress and inflammatory pathways (Fig. 8), may be a key advantage to pharmacologically target different aspects of the DMD pathology, through an epigenetic approach. For instance, IL6 transcript levels may be influenced both by p65 activation and by direct BRD4 regulation. Restoration of SIRT1 levels by JQ1 treatment may also potentially contribute to NFkB inactivation via p65 subunit deacetylation[69].

As shown by our in vitro model of oxidative stress induced by $H_2O_2$ treatment, Sirt1 protein levels are also vulnerable to ROS, which lead to Sirt1 post-translational modifications and protein degradation[69]. Notably, Sirt1 protein degradation is prevented by JQ1 treatment. Modulation of ROS metabolism by JQ1 administration also leads to Sirt1-dependent AMPK activation (Supplementary Figs. S4 and S5), which is consistent with data observed in the mdx skeletal muscle (Fig. 4d) and may provide a link to autophagy restoration, together with changes observed in Src/Akt axis activation.

Experiments in C2C12 myoblasts, in which oxidative stress was induced by $H_2O_2$ treatment, suggest that JQ1 ability to decrease ROS levels represents a more general capability, which can be extended to different sources of ROS production. Similar results were also recently reported in chondrocytes[70].

BRD4 level is enhanced in skeletal muscle of DMD patients and of mdx mice. Other changes in epigenetic factors have been observed in dystrophic muscle, such as augmented HDAC2 levels[26] and increased histone acetylation marks[27]. BRD4 stabilization may lead to altered transcription at specific chromatin domain characterized by acetylated histones, thus taking part to the altered genome-wide transcriptional program observed in the mdx muscle. Several factors, such as SPOP[71,72], DUB3[73], PIN1[74], and BRD4 post-translational modifications[75] have been involved in BRD4 protein stabilization in cancer models, and further investigation is warranted to understand the dynamics of BRD4 stabilization in the DMD and mdx muscle.

BET inhibitors appear to have different effects on C2C12 differentiation, according to dosage, selected chemical compound and timing of treatment[31,45,76]. The evidence indicating that in certain experimental conditions BET inhibitors can block differentiation has to be taken into careful account, particularly when considering potential translational applications with human experienced BET inhibitors. Nevertheless, our data suggest that decreased regeneration is associated to reduced damage and necrosis in the mdx muscle, and precedes the repeated cycles of degeneration and regeneration triggered by muscle necrosis. Thus, the positive effects observed in vivo are ascribed to a beneficial effect on myofiber integrity, which may delay satellite cell activation and the need for regeneration. In addition, we employed a relatively low dose of the BET inhibitor, which may preserve satellite cell proliferation and regeneration potential, as suggested by our in vitro data on isolated satellite cells.

Overall, our data demonstrate that BET inhibition holds the potential to counterbalance alterations of ROS metabolism in the dystrophic skeletal muscle, ameliorating myofibers physiology and muscle function. Preserving muscle integrity and improving performance by tuning ROS metabolism may also represent a promising approach in other conditions in which oxidative stress plays a pivotal role in skeletal muscle, such as in sarcopenia.

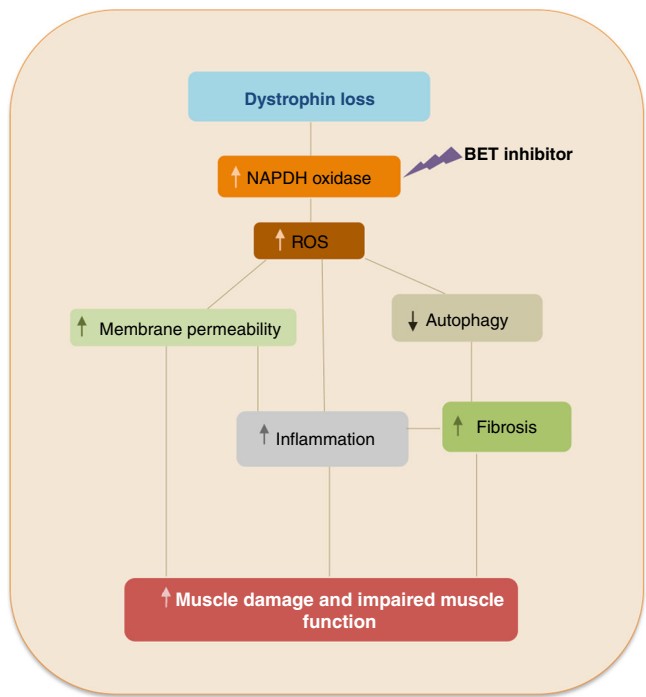

**Fig. 8 BET treatment and DMD physiopathology.** Working model for BET inhibitors effects on functional impairment in dystrophic muscle.

## Methods

**Animal study experimental design.** All procedures involving animal care or treatments were approved by the Italian Ministry of Health and performed in compliance with the guidelines of the Italian Ministry of Health (according to Legislative Decree 116/92), the Directive 2010/63/EU of the European Parliament and the Council on the protection of animals used for scientific purposes (Protocols 3/2014 and 791/2018). C57BL/10ScSn-*Dmdmdx*/J and control C57BL/10ScSnJ mice (Charles River, Italy) were housed in groups of five and maintained

under controlled temperature (20 ± 1 °C), humidity (55 ± 10%), and illumination (12/12 h light cycle with lights on at 07:30 am). Food and water were provided ad libitum. Tubes for tunneling and nesting materials (paper towels) were routinely placed in cages as environmental enrichment. Treatment was performed as in Segatto et al.[32]. For each litter, half of the mice were randomly allocated in the control group and half to the treatment group.

**Muscle morphological analysis and immunofluorescence**. In vivo morphological evaluation was performed on OCT frozen TA muscles. Transverse, 10-µm thick sections were cut by a cryostat and collected on Superfrost Plus slides (BioOptica). For each TA muscle ($n = 3$ per experimental group), a minimum of 20 sections were processed for hematoxylin-eosin (H&E) staining or Sirius red and dehydrated and mounted with Eukitt (Kindler GmbH & Co., Germany). Peripheral and centrally nucleated fibers imaging and analysis was not performed blinded to the treatment group.

For immunofluorescence analysis of TA muscles, 10-µm frozen sections were blocked with 10% Normal Goat Serum (NGS, Vector Laboratories) in PBS with 2,5% Triton-X 100, incubated with HCl for 20 min at RT and then incubated overnight at 4 °C with 8-OHdG antibodies.

Negative controls were performed omitting the primary antibodies. Sections were rinsed with PBS, incubated for 1 h at room temperature with Alexa 488 conjugated goat anti-rabbit IgG in 5% NGS PBS (1:500, Invitrogen Life Technologies, Carlsbad, CA, USA). Slides were mounted with Fluoroshield mounting medium with DAPI (Sigma Aldrich). Muscle slices were analyzed with a Leica CTR6000 microscope (Leica, Germany) equipped with Leica DFC360 camera (immunofluorescence visualization) and Leica DFC480 (bright field visualization). Images were captured using Leica Application Suite System and files were converted in Adobe photoshop CS5 format.

**Evans blue**. Evans blue dye (EBD) incorporation into necrotic/damaged fibers was assessed as previously reported[77]. Briefly, EBD (10 mg/ml in PBS) was dissolved and sterilized by using a 0.2-µm pore size filter. Subsequently, 0.05 ml/10 g body weight of dye solution was intraperitoneally administered to WT, vehicle- and JQ1-treated animals. The mice were then sacrificed 24 h after injection, and TA muscles were collected and immediately frozen in liquid nitrogen-cooled isopentane. Muscles were cut with a cryostat and analysis of EBD uptake was performed on 10-µm muscle cryosections. Muscle sections were fixed in cold acetone at −20 °C for 10 min, washed three times with PBS, coverslipped with aqueous mounting medium and evaluated by fluorescence microscopy. Imaging and analysis was not performed blinded to the treatment group.

**Treadmill test**. Mice were acclimatized to treadmill running with a 10-min run at a constant speed of 6 m/min. During the test sessions, mice were run at an initial speed of 6 m/min, and every 2 min speed was increased by 2 m/min until exhaustion. The first exercise test was used to set the baseline of each experimental group. JQ1 administration was started at the end of the training phase and was continued for the entire period of the exercise test (four weeks).

**Wire and inverted screen tests**. For wire test (or wire grip test) mice were tested at 1, 2, 3, and 4 weeks from the beginning of JQ1 administration. The animals were allowed to grasp by their four paws a 2-mm diameter metal wire, which was horizontally positioned 35 cm above a padded surface. The observers recorded the time spent on the wire, until the mice fell on the soft bedding. After each fall, the mice were allowed to rest for 1 minute. Each test consisted of three trials, and values derived from each trial were then averaged[78].

For the inverted screen test, the animals were tested at 1, 2, 3, and 4 weeks starting from the first day of treatment. Mice were placed in the center of a wire mesh screen, which was subsequently rotated to an inverted position over 2 s. The screen was held steadily 40 cm above thick soft bedding. The length of time until the animal fell on the padded surface was recorded. 60 s was considered the cut-off time in each trial[79].

Results derived from inverted screen tests were then analyzed by assigning the following scores: Falling between 1 and 10 s = 1; Falling between 11 and 25 s = 2; Falling between 26 and 60 s = 3; Falling after 60 s = 4.

**Isolation of EDL fibers and treatment**. EDL muscles were surgically isolated and incubated in DMEM containing 0.4% Collagenase type I at 37 °C for 1 h to release single fibers[80]. After 1 h in DMEM with 1% penicillin/streptomycin, fibers were transferred to a new dish in DMEM with 20% FBS, 1% chicken embryo extract and 1% penicillin/streptomycin and treated with JQ1 (200 nM) overnight and Trizol was added for RNA extraction.

**C2C12 treatments and immunofluorescence**. C2C12 cells (ATCC) were grown in DMEM high glucose with 20%FBS. Cells were pretreated with JQ1 (0.2 µM) for 24 h and $H_2O_2$ (250 µM) was added for 8 h or 24 h. For Sirt1 blockade, C2C12 myotubes were co-treated with JQ1 (0.2 µM) and Nicotinamide (NAM, 10 mM), 24 h prior to $H_2O_2$ stimulation. 24 h after $H_2O_2$ administration, C2C12 myotubes were harvested and processed for subsequent Western blot analysis. For 8OHdG

staining, immunofluorescence was performed as in Fenizia et al.[81], with one additional step of incubation with 2 N HCl for 20 min at RT, after cell fixation. The samples were examined with a Leica CTR6000 microscope (Leica, Germany) equipped with Leica DFC360 camera (immunofluorescence visualization) and Leica DFC480 (bright field visualization). Images were captured using Leica Application Suite System and files were converted in Adobe Photoshop CS5 format. DAPI was used for nuclear staining.

**Degradation assay**. Protein degradation assay was performed as in Segatto et al., 2014[82]. Briefly, C2C12 myotubes were treated with vehicle, $H_2O_2$ (250 µM) or co-treated with $H_2O_2$ (250 µM) and JQ1 (0.2 µM) as already reported. 24 h after $H_2O_2$ stimulation, cells were lysed in ice cold 0.01 M Tris-HCl (pH 7.4), 0.150 M sucrose. 30 µg of protein extraction was employed for each reaction. Samples were incubated at 37 °C, and the reaction was stopped by the addition of an equal volume of sample buffer (0.125 M Tris-HCl containing 10% SDS, protease inhibitor cocktail, pH 6.8), at different time points (2 h, 4 h, and 8 h). Samples were then boiled for 3 min and used for western blot analysis.

**siRNA C2C12 transfection**. C2C12 myoblasts were transfected in suspension with siBrd2 (0.1 µM), siBrd3 (0.1 µM), siBrd4 (0.1 µM) e siScramble (0.1 µM) (Supplementary Table 4), with Lipofectamine 2000 (Thermoscientific), according to manufacturer's instruction. After 24 h, cells were transfected again and after 24 h RNA was extracted.

**Western blot and qRT-PCR**. Muscle and protein extracts, immunoblot and RNA analysis was performed as in Segatto et al.[32]. Antibody and oligonucleotide lists are in Supplementary Tables 1 and 2, respectively. BRD4 antibody specificity was tested in C2C12 myoblasts silenced for BRD4 (Supplementary Fig. 1A). Quantitative Real Time PCR was performed using SYBR green IQ reagent (Bio-Rad Laboratories, Italy) with CFX Connect detection system (Bio-Rad Laboratories, Italy).

**Satellite cells isolation**. Satellite cells were isolated as in Proserpio et al.[31] and growth curve was performed as in Fenizia et al.[81].

**Analysis of intracellular ROS levels in C2C12 cells**. Cells were incubated with 5-(and-6)-chloromethyl-2′,7′-dichlorodihydrofluorescein diacetate acetyl ester (CMH2DCFDA) (Thermofisher) for 30 min at 37 °C and washed with PBS[4]. The green fluorescence intensity of the oxidized DCF probe per cell was quantified by Image J software and compared at the different time points.

**ChIP assay**. Chromatin isolated from muscles was subjected to ChIP assay as in Segatto et al.[32]. 100 mg of starting tissue was used for each antibody. Chromatin was sonicated to fragments length of approximately 0.5 Kb and immunoprecipitated with 3.5 µg of rabbit IgG or antibodies listed in Supplementary Table 1. ChIP primers are listed in Supplementary Table 3. Quantitative Real Time PCR was performed using SYBR green IQ reagent (Bio-Rad Laboratories, Italy) with CFX Connect detection system (Bio-Rad Laboratories, Italy).

**DMD immortalized cells and human tissues**. Approval from Ethics committee was obtained by the University of Milan. Human myoblast cells were immortalized by the Institut de Myologie (Pitié-Salpêtrière Hospital, Paris, France) (Supplementary Table 5). Cells were expanded in Skeletal muscle cell growth medium (Promocell C-23060) with Supplement mix (Promocell C-39365), 15% FBS, 1% L-glutamine, 1% gentamicine (Sigma G-1272) and 1% penicillin/streptomycin. Myoblast cells were treated with JQ1 (200 nM) for 24 h. DMD and control tissues were obtained from the AFM-Myobank (Paris) and tissues were processed as described for mouse muscles (Supplementary Table 6). The study design and conduct complied with all relevant regulations regarding the use of human study participants and was conducted in accordance to the criteria set by the Declaration of Helsinki. The participants, or their legal guardians, provided written informed consent.

**Statistics and reproducibility**. Data obtained from functional, morphological, western blot, and mRNA analysis are expressed as means ± SD (standard deviation). Measurements were taken from distinct samples, except for morphological analysis. All the biological replicates were checked for their normal distribution by using Shapiro–Wilk Test. When we compared 2 experimental groups we used unpaired $t$ test and when we compared 3 or more experimental groups we used one-way analysis of variance (ANOVA) followed by the Tukey's post hoc test. Statistical analysis for non-normal distributed data (inverted screen test, wire test) was performed by Kruskal–Wallis test followed by Dunns post hoc. Values of $p < 0.05$ were considered to indicate a significant difference. Experiments in Fig. 5g and Supplementary Fig. 1 were independently repeated 3 times; experiments shown in Supplementary Figs. 9B and 12E were independently repeated 2 times. Statistical analysis was performed using GRAPHPAD INSTAT3 (GraphPad, La Jolla, CA, USA) for Windows.

## Data availability

The data supporting the findings of this study are available from the corresponding author upon reasonable request. RNA-seq dataset for DMD and control muscles can be found as Supplementary Table in ref. [36] (https://stke.sciencemag.org/content/suppl/2012/08/03/5.236.ra56.DC1). Source data are provided with this paper.

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

## Acknowledgements

This work was supported by Telethon GGP13165, French Association for Myophaties (AFM-Telethon) and Duchenne Parent project (Italy), Cariplo 2017-0604 and AIRC IG 21353 grants to G.C. and MRC grant# MR/N010051/1 to P.F.R. Fittipaldi was supported by a FIRC fellowship. We are thankful to Dr. G. Strimpakos (CNR-Cell Biology and Neurobiology Institute, Rome) and the platform for immortalization of human cells at the Institut de Myologie (Paris) and Dr. C. Bragato (IRCSS Besta Milan) for technical support, to Dr. S. Vasseur and the Myobank-AFM at the Institute de Myologie, Paris (BB-0033-0012) for DMD tissues, to Dr. M. Mora and S. Gibertini (Besta Institute, Milan) for help with muscle sectioning, to Dr. J. Ervasti (Univ. of Minnesota) for the Tubulin6 antibody, and to Dr. G. Messina's group in our department for support with the treadmill tests. We are grateful to F. Penna (University of Turin, Italy) and Dr. D. Randazzo (NIAMS/NIH) for helpful discussions. This article is dedicated to the loving memory of Roberto Segatto.

## Author contributions

M.S. performed animal treatments, immunoblots, immunohistochemical, morphological, and functional analysis, helped with C2C12 experiments, designed experiments, analyzed data, and was involved in manuscript preparation. R.S. performed cell culture experiments, immunoblots, IF, and RNA analysis, 12 months old analysis. R.F. performed RNA analysis and helped with animal treatments. C.B. perfomed single fiber isolation experiments, L.N. performed immunoblots, M.C. generated DMD immortalized cell lines, P.F. provided JQ1 and contributed in experimental design and manuscript preparation. G.C. conceived the study and designed experiments, analyzed data and wrote the manuscript. All authors discussed the results and commented on the manuscript.

## Competing interests

The authors declare no competing interests.

## Additional information

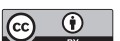

