## [Peer Review File · Nature Communications]

Reviewers' Comments:

Reviewer #1:

Remarks to the Author:

The authors address the role of bromodomain and extraterminal (BET) proteins, which are acetyl-chromatin binding factors, in the pathogenesis of muscular dystrophy. In muscle from the mdx mouse model of muscular dystrophy, expression of the BRD4 BET family member is elevated at the protein level. Treatment of mdx mice with the pan-BET inhibitor, JQ1, has various salutary effects, including increasing the number of intact skeletal muscle fibers and reducing the number of damaged fibers. This correlated with various effects that are predicted to be beneficial, such as reduced oxidative stress. In this regard, *in vitro* and *in vivo* studies revealed that JQ1 suppresses expression of NADPH oxidase subunits.

The ability of JQ1 to improve the muscular dystrophy phenotype is impressive and important. However, the manuscript lacks the detailed mechanistic insight and investigational rigor that is required for publication in *Nature Communications*.

Specific points

1. All *in vivo* studies employed three animals per group, which is unacceptable.
2. Additional studies should be performed to understand which BET family member(s) regulate the phenotype. This could be accomplished by knocking down expression of BRD2, BRD3 and BRD4 in the authors' cell culture models.
3. The ChIP-PCR findings in Fig. 5 are mislabeled on the y-axis. ChIP-PCR is not used to quantify mRNA expression.
4. The regulatory regions that were amplified in the ChIP-PCR study should be defined.
5. For immunoblotting, full-length gels with molecular weight markers should be shown.
6. In Fig. 4, C2C12 cells were treated with H₂O₂ and JQ1 for 24 hours, but in Fig. 5E, cells were pretreated with JQ1, then co-treated with H₂O₂ for 8 hours. The experimental conditions should be kept constant.
7. Since the idea is that JQ1 decreases oxidative stress in mdx mice, it would be better to pre-treat cells with H₂O₂, then co-treat with JQ1. Additionally, treatment with JQ1 alone should be included.
8. The authors suggest that SIRT1 accumulation and SRC phosphorylation are the links between oxidative stress and autophagy, although they did not dig deeper into this possibility. Additional immunoblots of LC3 and p62 for Fig. S3B would enhance the manuscript.

Reviewer #2:

Remarks to the Author:

In this report Segatto et al. demonstrate that the pharmacological inhibition of BET proteins has important structural and functional benefits on skeletal muscle quality of dystrophic mice as well as in primary myoblasts isolated from DMD donors. Authors use a variety of experimental methods however, there are important weaknesses that should be addressed.

- 1- There is a strong discrepancy between the mRNA and protein expression level of BRD24 in skeletal muscles of mdx mice. The authors suggest that this is possibly due to post-transcriptional

mechanisms. I think this is one of the major downsides of this report. The authors do not show any proof of this concept and do not provide any information about the specificity of the antibody used. The specificity of the anti-BRD4 must be evaluated in BRD4 KO mice or at least in BRD4 silenced C2C12 cells.

2- Another major downside of this report is the fully lack of information about JQ1. Is this compound selective for BRD4 or it has affinity also for the other BRD proteins? In this latter case, there is the risk that results shown in this report have been misinterpreted. In addition, there is no information about time (2 weeks) and dosage.

3- In some representative blots, the anti-vinculin or GAPDH antibody produces only one band while in other figures I see two bands. Can the authors explain why?

4- The authors do not provide any information about the bioinformatics analysis. Please provide all the necessary information about the consensus sequences, score, type of software used etc.

5- The authors state that BET proteins promote the transcriptional activity of NADPH oxidase sub genes. However, Chip assay is a useful tool to demonstrate the binding of transcription factors to DNA. It would be more straightforward to perform also luciferase assays.

6- The effect of JQ1 on muscle regeneration is very confusing. It would be a good idea to evaluate the effect of JQ1 on satellite cell proliferation and differentiation.

7- Since in mdx mice the onset is before 5 weeks and these mice do not show a decline in their regeneration capacity at early age, can the authors explain why they used 10 week-old mdx mice? What about the effect of JQ1 when the disease phenotype is more aggravated?

8- I noticed a complete lack of information about DMD donors. This needs to be clearly stated in the manuscript.

9- I have noted irregularities in the stats analysis. Authors stated that for experimental groups >2 they used ANOVA followed by Tukey's test. However, it is not mentioned in the manuscript if the biological replicates are normally distributed or not. If are not normally distributed they would need to be analysed via Kruskal-Wallis test with Dunn's post hoc (not Tukey) and be presented as box & whisker.

Reviewer #3:

Remarks to the Author:

This work from Segatto et al provides compelling evidence that BET inhibition with JQ1 ameliorates a number of the important pathological features of the mdx mouse model of Duchenne Muscular Dystrophy. The rationale and study design is well justified, the paper is clearly written and enjoyable to read, and the data overall appear robust and appropriately analyzed (with minor exceptions detailed below). In sum, the work reflects an impressive biochemical characterization, validated with functional rescue, and the overall result is both novel and significant. I have a few suggestions aimed at increasing the overall scope and relevance of the work, and some small text edits.

Major:

1. The relevance of this work would be significantly strengthened by a demonstration of BRD4 upregulation, or increased BRD4 occupancy of key targets such as Nox2, in human DMD tissue. The upregulation of BRD4 targets (such as Nox2) may be able to be simply mined from existing RNA sequencing data sets from DMD boys (such as in Khairallah et al., Science Signaling 2013). While the authors do show that JQ1 can reduce Nox2 transcript in immortalized DMD cells, a demonstration that BRD4 upregulation is a relevant feature beyond the mdx mouse would significantly strengthen the impact of this work.

Minor:

2. The authors conduct a fairly comprehensive biochemical characterization of many key dystrophic features ameliorated by JQ1 (oxidative stress, autophagy, inflammation, fibrosis). Given the relatively large number of recent reports linking Nox2-mediated ROS production to

microtubule abnormalities that contribute to DMD pathology (e.g. Randazzo et al., Hum Mol Genet 2019, Prosser et al., Science 2011, Loehr et al., Elife 2018, Nelson et al., Hum Mol Genet 2018, Kerr et al., Nature Communications 2015, Khairallah et al., Science Signaling 2013), it seems a missed opportunity to determine whether JQ1 also corrects microtubule abnormalities that contribute to disease pathology. This could be accomplished by a simple western blot of key markers of microtubule misregulation, such as expression of deetyrosinated-tubulin and TUBB6, and would provide further evidence that proximal changes in Nox2 contribute to the microtubule dysfunction well characterized in DMD.

3. For histological characterizations, for example of EBD and SHD imaging, there is no mention of the imaging or analysis being performed blinded to the treatment groups. These techniques are particularly prone to user bias, and positive results have historically been difficult to replicate when subjected to blinded analysis (there is a considerable history of this in the DMD field). These samples should be imaged and analyzed blindly, and if this is not possible, it should be stated as such in the methods.

4. Regarding the histology, the presence of centrally nucleated fibers is a classic hallmark of DMD pathology, and their correction a benchmark of pathological rescue. Thus it is odd that the authors do not quantify CNFs. This seems particularly important given the links to BET inhibition and differentiation/regeneration.

5. What is the definition of, or inclusion criteria for, an "intact" fiber as quantified in figure 1? Again, was this analysis performed blind?

6. The word "remarkably" is used in considerable excess (>10 times) to describe JQ1 effects. This degree of hyperbole throughout the manuscript is not needed, the robust results speak for themselves!

7. The authors state that the protection from oxidative stress "was not ascribed to the transcriptional activation of anti-oxidant genes", yet do not seem to probe for Nrf2, which regulates broad antioxidant proteins implicated in DMD pathology. This seems important to include, or at least explain, if the authors wish to make this claim.

8. Page 9 top, reference should be included for "as previously observed in mdx muscles"

9. Please edit page 12, "JQ1-treated mdx mice exhibited significantly increased resistance to fatigue in the treadmill test, and we observed a significant improvement in endurance."

10. Why not include the full time course of S. Fig 7 in Fig 6H? It seems odd to have two separate time courses, and the time course of relapse after JQ1 withdrawal is meaningful and warrants inclusion in the primary figures.

Authors response to reviewers' comments.

The authors thank the three reviewers for their efforts in reviewing our manuscript, for recognizing the novelty of our experimental work and the useful comments that we used to clarify and improve the manuscript.

We performed a substantial body of new experiments and below we present our point-to-point response, outlying modifications in the text, as well as in main or supplemental figures.

Novel data or changes in the text are shown in red.

Reviewer #1 (Remarks to the Author):

The authors address the role of bromodomain and extraterminal (BET) proteins, which are acetyl-chromatin binding factors, in the pathogenesis of muscular dystrophy. In muscle from the mdx mouse model of muscular dystrophy, expression of the BRD4 BET family member is elevated at the protein level. Treatment of mdx mice with the pan-BET inhibitor, JQ1, has various salutary effects, including increasing the number of intact skeletal muscle fibers and reducing the number of damaged fibers. This correlated with various effects that are predicted to be beneficial, such as reduced oxidative stress. In this regard, *in vitro* and *in vivo* studies revealed that JQ1 suppresses expression of NADPH oxidase subunits.

The ability of JQ1 to improve the muscular dystrophy phenotype is impressive and important. However, the manuscript lacks the detailed mechanistic insight and investigational rigor that is required for publication in Nature Communications.

Specific points

1. All *in vivo* studies employed three animals per group, which is unacceptable.

We employed more than 3 animals per group in our *in vivo* experiments, as described in figure legends. We usually used between 6-8 animals in immunoblots of Mdx-Veh and Mdx-JQ1 treated animals. We used an average of 7 animals per group for RNA qPCR analysis and used 3 animals per group in morphological evaluation of skeletal muscles, where multiple sections of each muscle were analyzed.

We employed 3 animals only for WT animals, which behaved in a very homogenous way.

2. Additional studies should be performed to understand which BET family member(s) regulate the phenotype. This could be accomplished by knocking down expression of BRD2, BRD3 and BRD4 in the authors' cell culture models.

We agreed with the reviewer and performed knockdown experiments with siRNAs against BRD2/BRD3/BRD4 and evaluated transcript levels of the NADPH oxidase subunits in the different conditions. These data are presented in Fig.5H.

Page 11-12:

"To define which BET protein plays a major role in NADPH oxidase subunits

modulation, we employed a siRNA approach in C2C12 myoblasts. BRD4 knockdown reduced the transcript levels of Nox2, Nox4, p47-phox and p67-phox, while BRD2 depletion didn't affect Nox4 mRNA but decreased Nox2, p47-phox and p67-phox expression. BRD3 did not influence NADPH subunits expression levels (Fig. 5H and Fig. S9A)."

3. The ChIP-PCR findings in Fig. 5 are mislabeled on the y-axis. ChIP-PCR is not used to quantify mRNA expression.

We thank you the reviewer for spotting this mistake. We corrected the y-axis in Fig. 5I, which represents the percentage of Input.

4. The regulatory regions that were amplified in the ChIP-PCR study should be defined.

We introduced a scheme indicating the position of the amplified regions in the genes and explained in the text that we derived these regulatory regions by our previously published ChIP-seq data and from previous reports (Page 12).

"Because of BRD2 and BRD4 ability to modulate NADPH oxidase subunits, we performed ChIP assays for these two BET proteins in skeletal muscles of control, vehicle- and JQ1-treated mdx mice. We amplified chromatin regulatory regions that we previously found to be occupied by BRD4 in TA of control mice by ChIP-seq assays,³² which were shown to include previously described regulatory regions⁵⁵⁻⁶⁰."

5. For immunoblotting, full-length gels with molecular weight markers should be shown.

Representative gels for main and supplemental figures, with molecular weight markers, are reported in the Data Source file.

6. In Fig. 4, C2C12 cells were treated with H₂O₂ and JQ1 for 24 hours, but in Fig. 5E, cells were pretreated with JQ1, then co-treated with H₂O₂ for 8 hours. The experimental conditions should be kept constant.

Some piece of information was missing in figure legend 4. In the experiments shown in Fig. 4, we also pre-treated cells with JQ1 for 24 hr, as in Fig.5, and then we co-treated with H₂O₂ for 8 hours. While the time of JQ1 pre-treatment was the same in the two figures (24hr), we employed different H₂O₂ treatment time (8 versus 24 hours). The rationale for this difference in H₂O₂ treatment was that in figure 5E we were looking at the RNA and in Fig. 4 at the protein levels: we reasoned that RNA changes precede protein alterations. We have changed figure legend 4B-D accordingly, adding the information regarding pre-treatment.

Page 27: "Cells were pretreated with JQ1 (200nM) for 24 hr and then stimulated with H₂O₂ for 24 hr".

7. Since the idea is that JQ1 decreases oxidative stress in mdx mice, it would be better to pre-treat cells with H₂O₂, then co-treat with JQ1. Additionally, treatment with JQ1 alone should be included.

We performed these experiments and reported these data in Supplementary Fig. S5.

We have found that our initial data were confirmed in this setting.

These figures are now described on page 9-10:

Page 9-10. "To test whether JQ1 is effective when oxidative stress is already established,

we administered JQ1 after treating C2C12 myotubes with H₂O₂ for 2 hours, a sufficient time to test whether JQ1 is effective when oxidative stress is already established, we administered JQ1 after treating C2C12 myotubes with H₂O₂ for 2 hours, a sufficient time to induce oxidative stress in C2C12 cells (Fig. S5A). The modulation of Sirt1, p62, LC3 as well as of phosphorylated AKT, AMPK and Ulk1 were similar to the one obtained when cells were pretreated with JQ1, followed by H₂O₂ stimulation (Fig. S5B-C). In addition, JQ1 treatment alone did not affect p-AKT and p62 levels, but it increased Sirt1, lipidated LC3 levels, and AMPK and Ulk1 phosphorylation (Fig. S5B-D). In this experimental setting, we confirmed that NAM treatment prevented AMPK activation and ULK phosphorylation (Fig. S5C)."

8. The authors suggest that SIRT1 accumulation and SRC phosphorylation are the links between oxidative stress and autophagy, although they did not dig deeper into this possibility. Additional immunoblots of LC3 and p62 for Fig. S3B would enhance the manuscript.

The link between SRC phosphorylation, oxidative stress and autophagy in the mdx mouse was previously reported (Pal et al., 2014).

We performed the suggested experiments and included these results in Supplemental Fig. 5D. We discussed these results on page 10.

"Moreover, Sirt1 pharmacological blockade hindered LC3 accumulation and p62 downregulation in cells in which oxidative stress was induced by H₂O₂ followed by JQ1 treatment (Fig. S5D), further supporting the idea that Sirt1 plays a key role in AMPK activation and autophagy regulation."

Reviewer #2 (Remarks to the Author):

In this report Segatto et al. demonstrate that the pharmacological inhibition of BET proteins has important structural and functional benefits on skeletal muscle quality of dystrophic mice as well as in primary myoblasts isolated from DMD donors. Authors use a variety of experimental methods however, there are important weaknesses that should be addressed.

1- There is a strong discrepancy between the mRNA and protein expression level of BRD24 in skeletal muscles of mdx mice. The authors suggest that this is possibly due to post-transcriptional mechanisms. I think this is one of the major downsides of this report. The authors do not show any proof of this concept and do not provide any information about the specificity of the antibody used. The specificity of the anti-BRD4 must be evaluated in BRD4 KO mice or at least in BRD4 silenced C2C12 cells.

We have tested the specificity of the anti-BRD4 antibody in BRD4 silenced cells and reported this data in Supplemental Fig.S1 (Material and Methods section).

Page 23: "BRD4 antibody specificity was tested in C2C12 myoblasts silenced for BRD4 (Fig. S1A)."

We agree that the difference between mRNA and protein level is an interesting but open question. Data found on DMD muscles and included in Fig. 1C,D are pointing to a relevant role for this stabilization also in the human pathology, where BRD4 protein

levels are higher in DMD muscles but transcript levels are comparable to healthy donors, in independently published RNA-Seq datasets. We are working on the mechanisms underlying BRD4 stabilization, which will require several efforts to define possible post-translational modifications affecting protein degradation and/or to establish which E2/E3 complex is involved in BRD4 degradation and its modulation in the mdx muscle. Since the available tools to dissect these molecular mechanisms are still limited (for instance antibodies for BRD4 post-translational modifications), we will include these studies in a future manuscript.

2- Another major downside of this report is the fully lack of information about JQ1. Is this compound selective for BRD4 or it has affinity also for the other BRD proteins? In this latter case, there is the risk that results shown in this report have been misinterpreted. In addition, there is no information about time (2 weeks) and dosage.

JQ1 binds to both BRD3 and BRD4 with a similar Kd and with higher Kd to BRD2 (Filippakopoulos et al., 2010). Interestingly, BRD2 and BRD4 were shown to regulate transcription in concert in several contexts (Cheung K. et al, 2017; Frnandez-Alonso R. et al., 2017).

More targeted loss of function experiments, through siRNA transfection, were used to address the ability of BET proteins to regulate specific targets.

These data were included in Fig. 5H and S9A and were described on pages 11-12.

“JQ1 binds with different affinity the BET proteins, thus being capable to displace BRD2/BRD3/BRD4 from chromatin⁵³. To define which BET proteins play a major role in NADPH oxidase subunits modulation, we employed a siRNA approach in C2C12 myoblasts. BRD4 knockdown reduced the transcript levels of Nox2, Nox4, p47-phox and p67-phox, while BRD2 depletion didn't affect Nox4 mRNA but decreased Nox2, p47-phox and p67-phox expression (Fig. 5H and Fig. S9A).”

We also introduced more information about JQ1 chronic treatment.

Page 6: “We daily treated 10-week-old mdx mice with JQ1 (20 mg/kg per day) by intraperitoneal injection”

3- In some representative blots, the anti-vinculin or GADPH antibody produces only one band while in other figures I see two bands. Can the authors explain why?

Vinculin can display a main band and a fainter upper band (metavinculin), while GAPDH can display a fainter lower band in certain cell types (photo 4, <https://www.abcam.com/gapdh-antibody-6c5-loading-control-ab8245.html>). In some of our representative gels the second fainter band was not present because of the thicker gel concentration, running conditions or exposure settings. Representative western blot are now included in Data source.

4- The authors do not provide any information about the bioinformatics analysis. Please provide all the necessary information about the consensus sequences, score, type of software used etc.

We did not show ChIP-Seq data and thus we did not provide this information.

5- The authors state that BET proteins promote the transcriptional activity of NADPH

oxidase sub genes. However, Chip assay is a useful tool to demonstrate the binding of transcription factors to DNA. It would be more straightforward to perform also luciferase assays.

BRD4 doesn't directly bind to DNA and it requires histones for its anchoring to the chromatin. Luciferase assays are useful means to show the ability of transcription factors to promote transcription, once bound to their consensus site; their relevance is less straightforward with chromatin factors. In particular BRD4 requires acetylated histones for recruitment to DNA, which may not be properly assembled on plasmid DNA in a transient transfection assay. ChIP is a more direct and less artificial mean to prove BRD4 chromatin occupancy to genomic regions.

We do agree with the reviewer that chromatin association doesn't necessarily imply transcription, but the siRNA assays we introduced in fig. 5H show that BRD2 and BRD4 depletion do influence NADPH subunit levels. Therefore, we changed the text on page 12 accordingly:

“Overall, these findings revealed that BRD4 and BRD2 occupy the NADPH oxidase subunits regulatory genes in the mdx muscle and modulate their transcription.”

6- The effect of JQ1 on muscle regeneration is very confusing. It would be a good idea to evaluate the effect of JQ1 on satellite cell proliferation and differentiation.

We isolated satellite cells and evaluated JQ1 effects on proliferation and differentiation.

These data are now included in Supplemental figure S10 and are described on page 14:

“In vitro, JQ1 (200nM) treatment of satellite cells did not prevent their ability to differentiate, nor significantly decreased their proliferation rate (Fig. S12D and E).”

7- Since in mdx mice the onset is before 5 weeks and these mice do not show a decline in their regeneration capacity at early age, can the authors explain why they used 10 week-old mdx mice? What about the effect of JQ1 when the disease phenotype is more aggravated?

We employed 10 weeks old animals to depict a disease stage in which the muscle is still plastic, but it also displays several concurrent hallmarks of the pathology. For example, we were able to study fibrosis at this stage, since it starts to manifest around 12 weeks. Importantly, we were also interested in autophagy, which is still active at 6 weeks and get less functional later on (Fiacco et al., 2016).

In addition, Nox2 protein is upregulated at 5-9 weeks of age in the mdx mouse (Whitehead et al., 2010), but not in younger animals.

Another aspect of interest was inflammation, which peaks around 4-8 weeks but it is still present at 10-12 weeks. Therefore, we selected a timepoint in which we were able to dissect different aspects of the disease pathophysiology, of the adult mouse.

We do agree that it is interesting to further characterize JQ1 effects with a more aggravated phenotype. We treated mdx mice at 11 month of age for 4 weeks and focused out attention on overall morphology, inflammation, fibrosis and NADPH oxidase subunits transcription level. The longer treatment time was employed because we reasoned that certain pathological features, such as fibrosis, would be more difficult to

challenge in older mice.

These data are presented in Supplemental Fig. S11 and described on page 13-14:

“To evaluate the impact of JQ1 treatment in older animals, we daily treated 11 months old mdx mice with JQ1 (20mg/ml/day) by intraperitoneal injection, for 4 weeks. The longer treatment time was employed because we reasoned that certain pathological features, such as fibrosis, would be more difficult to challenge in older mice. At this stage of the disease progression, JQ1 administration led to a reduction in the transcript levels of inflammatory markers TNF α and IL6 (Fig. S11A), which was paralleled by a decrease in the levels of CD45 and F4/80 proteins (Fig. S11B), as well as of inflammatory infiltrate (Fig. S11C). BET blockade led to a trend towards increasing the number of peripheral nucleated fibers and reducing the centrally nucleated fibers, although not significantly (Fig. S11D). Fibrosis was reduced in 12 months old JQ1-treated mdx TAs, as shown by Sirius red staining (Fig. S11E). Transcript levels of NADPH oxidase subunits and collagen 1 α 1 were also reduced following JQ1 administration (Fig. S11F and G). Overall, these results show that, in the mdx mouse model, JQ1 treatment has a beneficial impact also when the disease phenotype is aggravated.”

8- I noticed a complete lack of information about DMD donors. This needs to be clearly stated in the manuscript.

We included this information in Supplemental materials, in supplemental Methods, page 1.

Donor myoblasts for immortalized cell lines

Dystrophin gene mutation	Reference	Muscle	Age
mutation stop exon 59: c.8713C>T, p.Arg2905X	AB1023DMD11Q	Quadriceps	11 y
Del 45-50	KM1315DMD14PV	Paravertebral	14 y
Duplication exon 10-11	KM1316DMD14D	Dorsal	14 y

9- I have noted irregularities in the stats analysis. Authors stated that for experimental groups >2 they used ANOVA followed by Tukey's test. However, it is not mentioned in the manuscript if the biological replicates are normally distributed or not. If are not normally distributed they would need to be analysed via Kruskal-Wallis test with Dunn's post hoc (not Tukey) and be presented as box & whisker.

Normality distribution was checked through the Sapiro Wilk Test, as shown in the excel file enclosed. Statistical analysis for non-normal distributed data (inverted screen test, wire test) was performed by Kruskal-Wallis test followed by Dunns post hoc and this information was included in the material and methods.

Reviewer #3 (Remarks to the Author):

This work from Segatto et al provides compelling evidence that BET inhibition with JQ1 ameliorates a number of the important pathological features of the mdx mouse model of

Duchenne Muscular Dystrophy. The rationale and study design is well justified, the paper is clearly written and enjoyable to read, and the data overall appear robust and appropriately analyzed (with minor exceptions detailed below). In sum, the work reflects an impressive biochemical characterization, validated with functional rescue, and the overall result is both novel and significant. I have a few suggestions aimed at increasing the overall scope and relevance of the work, and some small text edits.

Major:

1. The relevance of this work would be significantly strengthened by a demonstration of BRD4 upregulation, or increased BRD4 occupancy of key targets such as Nox2, in human DMD tissue. The upregulation of BRD4 targets (such as Nox2) may be able to be simply mined from existing RNA sequencing data sets from DMD boys (such as in Khairallah et al., Science Signaling 2013). While the authors do show that JQ1 can reduce Nox2 transcript in immortalized DMD cells, a demonstration that BRD4 upregulation is a relevant feature beyond the mdx mouse would significantly strengthen the impact of this work.

We agreed with the reviewer that showing BRD4 levels in DMD patients is a relevant point, and we made several attempts to obtain DMD tissues from European and US biobanks. We were able to obtain 4 DMD and 2 control samples from the AFM-Myobank (Paris) and data show that BRD4 levels increase in DMD muscles, despite the limited number of control age-matched muscles. These data are reported in fig. 1C.

We also confirmed a significant decrease of Nox2, p47-phox and p67-phox transcripts in RNA-seq data of DMD vs control muscles, published in Khairallah et al. (Khairallah et al., Science Signaling 2012). These results are reported in Fig. 5E. From the same dataset we also found that BRD4 transcript levels are not significantly different in healthy and DMD donors (Fig 1D).

Page 5: “ We next analyzed BRD4 levels in DMD muscle samples, and found that BRD4 protein was higher in muscles of DMD patients than in aged matched controls (Fig. 1C). We therefore interrogated RNA-Seq results published by Khairallah et al.³⁶ and found that BET transcript levels do not significantly change in DMD muscles (Fig.1D).”

Minor:

2. The authors conduct a fairly comprehensive biochemical characterization of many key dystrophic features ameliorated by JQ1 (oxidative stress, autophagy, inflammation, fibrosis). Given the relatively large number of recent reports linking Nox2-mediated ROS production to microtubule abnormalities that contribute to DMD pathology (e.g. Randazzo et al., Hum Mol Genet 2019, Prosser et al., Science 2011, Loehr et al., Elife 2018, Nelson et al., Hum Mol Genet 2018, Kerr et al., Nature Communications 2015, Khairallah et al., Science Signaling 2013), it seems a missed opportunity to determine whether JQ1 also corrects microtubule abnormalities that contribute to disease pathology. This could be accomplished by a simple western blot of key markers of microtubule misregulation, such as expression of deetyrosinated-tubulin and TUBB6, and would provide further evidence that proximal changes in Nox2 contribute to the microtubule dysfunction well characterized in DMD.

We thank the reviewer for this interesting suggestion. We performed immunoblots with

detyrosinated-tubulin, alpha tubulin and Tubulin 6. Tubulin 6 results are particularly striking, because of its strong decrease following JQ1 treatment. These data are now reported in fig. 6H, and described on page 14 in the text.

“In DMD muscles, dystrophin absence alters the cytoskeleton, which results as a disorganized net of denser microtubules. Since the microtubules network conveys mechanotransduction signals to Nox2-dependent enhancement of ROS^{17,36,62-64} in adult mdx muscles, we asked whether JQ1 treatment was able to correct microtubules anomalies that contribute to the DMD pathology. We confirmed that total and de-tyrosinated alpha-tubulin is increased in adult mdx muscles, and we found that JQ1 treatment decreased both alpha-tubulin and de-tyrosinated tubulin (Fig. 6H). Tubulin6 protein significantly increased in adult mdx TAs when compared to control animals^{63,64}, and JQ1 reduced its levels to the ones of control mice (Fig. 6H).”

3. For histological characterizations, for example of EBD and SHD imaging, there is no mention of the imaging or analysis being performed blinded to the treatment groups. These techniques are particularly prone to user bias, and positive results have historically been difficult to replicate when subjected to blinded analysis (there is a considerable history of this in the DMD field). These samples should be imaged and analyzed blindly, and if this is not possible, it should be stated as such in the methods.

The imaging or analysis wasn't performed blinded, we added this statement in the methods, page 20.

4. Regarding the histology, the presence of centrally nucleated fibers is a classic hallmark of DMD pathology, and their correction a benchmark of pathological rescue. Thus it is odd that the authors do not quantify CNFs. This seems particularly important given the links to BET inhibition and differentiation/regeneration.

We performed this analysis and introduced a graph in figure 2B and in figure S11D, in which 12 months old animals were used.

These data are described on page 6 and 14.

Page 6: “JQ1 administration significantly increased the number of peripherally nucleated fibers and decreased the centrally nucleated fibers (Fig. 2B)”

Page 13-14: “BET blockade led to a trend towards increasing the number of peripheral nucleated fibers and reducing the centrally nucleated fibers, although not significantly (Fig. S11D).”

5. What is the definition of, or inclusion criteria for, an “intact” fiber as quantified in figure 1? Again, was this analysis performed blind?

Since it was not very clear, we changed the label of intact fiber to “peripheral nucleated fibers”. This definition is more intuitive also in comparison with the closely related centrally nucleated fibers. This count was also not blinded and it stated in the methods section (page 19).

“Peripheral and centrally nucleated fibers imaging and analysis wasn't performed blinded to the treatment group.”

6. The word “remarkably” is used in considerable excess (>10 times) to describe JQ1 effects. This degree of hyperbole throughout the manuscript is not needed, the robust results speak for themselves!

We deleted the word “remarkably” in the text, except once.

7. The authors state that the protection from oxidative stress “was not ascribed to the transcriptional activation of anti-oxidant genes”, yet do not seem to probe for Nrf2, which regulates broad antioxidant proteins implicated in DMD pathology. This seems important to include, or at least explain, if the authors wish to make this claim.

We did analyze Nrf2 targets in Fig. S6, and now introduced an immunoblot showing that Nrf2 doesn't significant change following JQ1 treatment (S7B). Nrf2 transcript is also unaffected by JQ1 administration (S7A). These data are described on page 11.

“In the mdx muscle, JQ1 administration did not affect mRNA and protein levels of Nrf2, a transcription factor that plays a key role in the antioxidant response pathway (Fig. S7A, B). Likewise, JQ1 treatment did not alter the transcriptional regulation of Nrf2 targets, Hmox1, Gclm and Gclc (Fig. S7C), suggesting that restoration of ROS metabolism was not ascribed to the transcriptional activation of anti-oxidant genes.”

8. Page 9 top, reference should be included for “as previously observed in mdx muscles”

This wasn't a previously published result, but we refer at our data presented in Fig.4 of this manuscript. We rephrased the sentence as:

Page 9: “Nevertheless, JQ1/H₂O₂ co-treatment restored LC3II abundance and reduced p62 levels, as observed for mdx muscles in Fig. 4C.”

9. Please edit page 12, “JQ1-treated mdx mice exhibited significantly increased resistance to fatigue in the treadmill test, and we observed a significant improvement in endurance.”

This sentence is now on pages 15. We did edit this phrase as follow:

“In agreement with the overall reduced muscle damage, JQ1-treated mdx mice significantly increased resistance to fatigue in the treadmill test, and we observed a substantial amelioration in endurance.”

10. Why not include the full time course of S. Fig 7 in Fig 6H? It seems odd to have two separate time courses, and the time course of relapse after JQ1 withdrawal is meaningful and warrants inclusion in the primary figures.

We agreed with the reviewer to reunite time courses. Since Fig. 6 was very crowded, particularly after adding the tubulin immunoblots, we decided to move time courses in a separate figure, Fig 7.

Reviewers' Comments:

Reviewer #1:

Remarks to the Author:

The authors have addressed most of my concerns. Two issues need to be addressed.

1. Please show immunoblots of BRD2, BRD3 and BRD4 for the siRNA study in Fig. 5.
2. I don't see full-length blots with MW markers in the Data Source file.

Reviewer #2:

Remarks to the Author:

Dear authors,

This manuscript has been revised and most of the input that I gave in the first review has been considered. I thank the authors.

However, I still think worth evaluating the quality of western blot experiments. This is especially if a major conclusion is "BET transcript levels do not significantly change in mdx and DMD muscles". In fig. 1 is shown only TA muscle. Why the authors did not explore the BRD4 expression in other muscle types? And also, the experiment shown in S1A is poor. Why is not quantified? Why the GAPDH signal is so weak in C2C12?

Again, JQ1. The authors show convincing experiments in C2C12, but they don't provide information about the expression levels of the three BRD isoforms. Which is the most abundant one?

Moreover, since TA is a fast glycolytic muscle, I wonder whether c2C12 cells originate from the same muscle type.

It's unclear to me why some figure 5 C D, H doesn't include the error bar in control groups.

It is essential to include statements in the Methods as to how randomization and blinding were performed

In my opinion, the work still needs some revisions

Reviewer #3:

Remarks to the Author:

The authors appropriately addressed my one major and few minor concerns with new experiments, analysis, and contextualization that strengthen the manuscript. In the future I would advise the authors to perform blinded analysis of DMD (and other) phenotypes whenever possible, as many of these assays are particularly subject to unconscious bias. That said, the authors are now transparent about the analysis performed here, which is important. And while of course there are still open questions about mechanisms of BRD4 stabilization and action, the work here is expansive in scope and represents a significant advance, and I have no remaining major concerns.

First, we would like to thank again the reviewers for their suggestions and their useful comments that have contributed to improving our manuscript.

Reviewer #1 (Remarks to the Author):

The authors have addressed most of my concerns. Two issues need to be addressed.

1. Please show immunoblots of BRD2, BRD3 and BRD4 for the siRNA study in Fig. 5.

We included immunoblots of BRD2, BRD3 and BRD4 in Supplemental Fig. 9B.

2. I don't see full-length blots with MW markers in the Data Source file.

We are sorry that the Reviewer wasn't able to see blots in the Data Source file. After uploading the current version of the Data Source file, we have contacted an editorial assistant at Nature Communications and made sure that the images are visible.

Reviewer #2 (Remarks to the Author):

Dear authors,

This manuscript has been revised and most of the input that I gave in the first review has been considered. I thank the authors.

However, I still think worth evaluating the quality of western blot experiments. This is especially if a major conclusion is "BET transcript levels do not significantly change in mdx and DMD muscles ". In fig. 1 is shown only TA muscle. Why the authors did not explore the BRD4 expression in other muscle types? And also, the experiment shown in S1A is poor. Why is not quantified? Why the GAPDH signal is so weak in C2C12?

We have quantified BRD4 reduction in Fig S1A and showed that BRD4 protein level is reduced by 84% following siRNA transfection in C2C12 cells.

GAPDH signal was weak because we selected one of the shortest exposures from the acquisition system (Chemidoc Biorad). We now changed this image and used a longer exposure. Also, controls for BRD siRNA experiments are now also shown as immunoblots (Fig. S9B) and in this experiment as well, BRD4 levels was reduced by 84% after siRNA transfection, further supporting the specificity of the band.

In addition, we have also employed a different BRD4 antibody (Santa Cruz sc-27976) in the past, and obtained comparable results regarding BRD4 stabilization in skeletal muscle. Besides, the antibody we used (raised between aa 1312 and 1362) is widely employed and cited, as it is working in a wide range of applications, spanning from CHIP to paraffin-fixed IHC, and represents one of the best choices for BRD4.

We focused our analysis on TA since this muscle is the most widely used as a prototype model system for DMD research. TA is one of the most active lower leg muscles and, differently from other mdx muscles, is highly susceptible to contraction-induced injury, thus representing an interesting model to study the secondary effects of dystrophinopathy (Dellorusso et al., 2001; Carberry et al., 2012). TA muscle is also extremely well characterized because of different experimental advantages, as it is located in an ideal position for functional testing, and methodological manipulation. Indeed, it is extensively studied for evaluating the efficacy of pharmacological therapies (Gilbert et al., 1999; Greelish et al., 1999; Ebihara et al., 2000; Ferrer et al., 2000; Wakefield et al., 2000; Byun et al., 2001) and of gene therapy interventions (Dellorusso et al., 2001; Sharp et al., 2010).

In addition to being extensively used to assess efficacy of pharmacological treatment, TA is a well-accepted model in which one can study autophagy and oxidative stress at the same time, which were processes of interest for our studies (De Palma et al., 2012; Whitehead et al., 2010; Spitali et al., 2013; Pal et al., 2014; Abou Samra et al., 2015).

Again, JQ1. The authors show convincing experiments in C2C12, but they don't provide information about the expression levels of the three BRD isoforms. Which is the most abundant one?

Since BRD2, BRD3 and BRD4 are not isoforms, it is not possible to address the relative protein levels by western blots, for instance using a single antibody. Thus, it is not possible to easily assess the relative protein cell concentration for BRD2/BRD3/BRD4. To provide the reviewer with quantitative data, we have analyzed transcript levels using publically available dataset for RNA-seq performed in C2C12 myoblasts. We have found that the most abundant transcript is BRD2, followed by BRD4 and the least expressed is BRD3.

We added this useful piece of information in the text (page 8), while introducing the C2C12 model.

To further investigate JQ1 impact on ROS metabolism in the mdx muscle, we asked whether ROS levels were affected by JQ1 treatment in C2C12 cells, where previously published RNA-seq datasets show that BRD2/3/4 are highly expressed, with BRD2 transcript being the most abundant followed by BRD4 and then BRD3 ^{23,44-46}.

Moreover, since TA is a fast glycolytic muscle, I wonder whether c2C12 cells originate from the same muscle type.

C2C12 cells originate from satellite cells, which were isolated from the thigh of a 2-month old female mouse (Saxel and Yaffe, 1977). The muscle of origin has the ability to influence the resident skeletal muscle stem cells metabolic state (Cerletti et al., 2012; Rocheteau et al., 2012, Pala et al., 2018; Ryall et al., 2016, Motohashi 2019), but after many years of in vitro culture it is difficult to say whether the abundance of fiber II versus fiber I type can be influenced by the

muscle of origin, in C2C12 cells. Data in the literature, though, suggest that, when compared to other human and mouse myoblast cell lines, C2C12 cells express a higher level of Myh1 and Myh4 proteins, associated with type II glycolytic fibers (Schiaffino and Reggiani, *Physiol Rev* 2011; Abdelmoez AM et al., *Am Journal of Cell Physiology* 2020).

It's unclear to me why some figure 5 C D, H doesn't include the error bar in control groups.

Independent experiments performed on immortalized DMD cells and siRNA interfered cells were collected at different times and analyzed together. Thus, we have re-analyzed the data and included the error bar in control groups of figure 5 D and 5H.

Independent experiments on isolated myofibers, instead, were conducted and analyzed separately, since these experiments are more complex and require longer times for planning and setting, because it requires tissues harvested from animals. Thus, we are unable to include error bar in the control groups of Fig. 5C because, differently from figure 5 D and H, we performed 3 independent experiments that were separately analyzed over time in independent qRT-PCR runs. In each experiment, JQ1-treated myotubes were compared to the respective control that was set as 1. We hope the reviewer may endorse the way we represent the relative fold change in this experiment, which is commonly reported in several articles, including many published in *Nature Communications* (Milan et al., 2015; Teng et al., 2016; Bhaskaran et al., 2019; Ertay et al., 2020)

It is essential to include statements in the Methods as to how randomization and blinding were performed

We added this statement in the Methods section (page 19):

“For each litter, half of the mice were randomly allocated in the control group and half to the treatment group.”

In my opinion, the work still needs some revisions

Reviewer #3 (Remarks to the Author):

The authors appropriately addressed my one major and few minor concerns with new experiments, analysis, and contextualization that strengthen the manuscript. In the future I would advise the authors to perform blinded analysis of DMD (and other) phenotypes whenever possible, as many of these assays are particularly subject to unconscious bias. That said, the authors are now transparent about the analysis performed here, which is important. And while of course there are still open questions about mechanisms of BRD4 stabilization and action, the work here is expansive in scope and represents a significant advance, and I have no

remaining major concerns.

We thank the reviewer and appreciate his/her comments. We agree on the advice to perform blinded analysis for our future experiments on DMD animal models, and we will follow this recommendation.

Reviewers' Comments:

Reviewer #1:

Remarks to the Author:

1.The cropped blots for sFig. 9B do not appear to match what is shown in the Data source file for the full-length blots

2.Please explain why BRD4 migrates above the 250 kDa marker in sFig. 1A and well below the marker in sFig. 9B (full-length blots in the Data source file)

3.sFig. 9B full-length blots are not clearly labeled

Reviewer #2:

Remarks to the Author:

The authors addressed all of my concerns. I think that the authors did all their best.

Response to Reviewer 1

1. The cropped blots for sFig. 9B do not appear to match what is shown in the Data source file for the full-length blots

We apologize for the mistake in the final version of R2. We have now corrected the blot in Supplementary Fig 9B.

2. Please explain why BRD4 migrates above the 250 kDa marker in sFig. 1A and well below the marker in sFig. 9B (full-length blots in the Data source file)

The corrected BRD4 shown in Supplementary fig 9B migrates at the same height as Supplementary Fig. S1A.

3. sFig. 9B full-length blots are not clearly labeled.

We labeled blots in Supplementary Fig. 9B in a clearer way.

Response to editorial requests.

We reported our corrections in Extended comments report file. We choose not to move fig 7 in supplementary because it is a very relevant figure to show functional recovery in the mdx muscle.